# A Multi-resolution Dataset of Self-consistent Cloth Drapes for Physics-based Upsampling

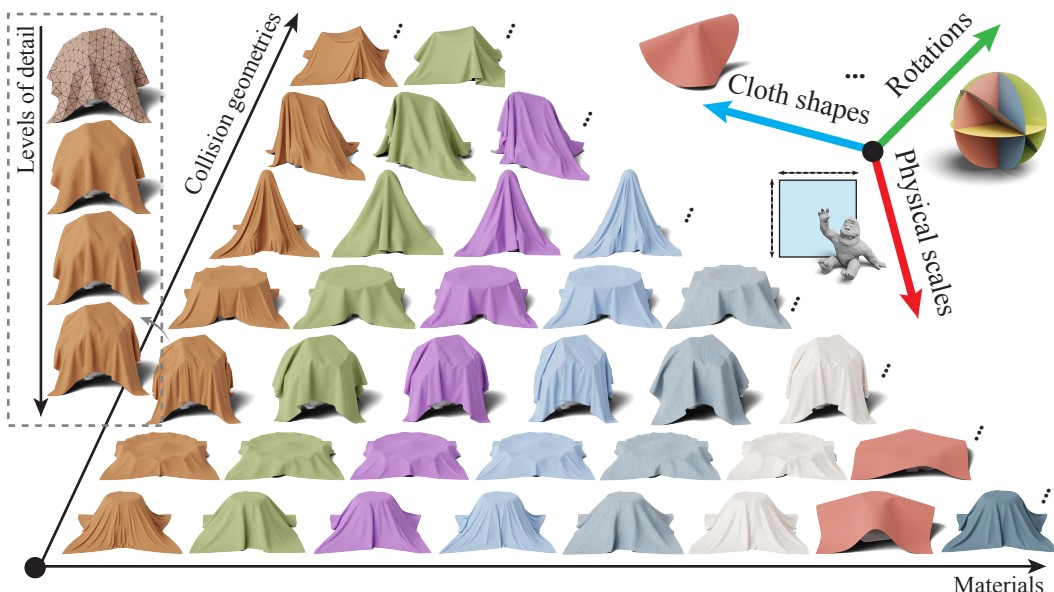

Figure 1: Our dataset provides over one million multi-resolution high-fidelity cloth drape geometries parameterized with wide variation, over six key differentiating features in cloth simulation: 1) shapes; 2) materials; 3) initial drape configurations (rotations); 4) cloth physical sizes and thicknesses; 5) collider geometries; and 6) collider sizes. We display here a representative subset of our dataset's geometries along these varying axes. For each simulated drape we provide a set of *self-consistent* increasing resolution simulation meshes with high-quality, artifact-free and intersection-free modeling of simulated cloth physics across all resolutions. Both consistency across resolutions and quality at the finest levels of resolution are not possible in prior SOTA cloth simulators (both industrial and academic), making it uniquely suited for advancing research and analysis in the development of physics-based upsampling methods.

## ABSTRACT

The high-fidelity simulation of draped cloth is a crucial tool across a wide range of applications that span the gamut from immersive virtual experiences to accurate digital modeling. However, capturing the finely detailed folds, creases, contacts and wrinkles of a cloth at equilibrium requires expensive, high-resolution simulation. To side-step these intensive computation requirements, data-driven methods have long attempted to directly upsample cheap-to-generate, low-resolution, coarse-cloth drapes with rich, physically realistic cloth-drape details. However, progress in these "physics upsampling" methods is significantly stymied by the lack of suitable data to capture both the intricate details of cloth physics and, just as important, to provide consistent, artifact-free, multi-resolution models of draping to learn the changes and correspondences across resolutions. Existing cloth simulators (both industrial and academic) generally fail to capture accurate draping behavior of real-world materials, lack the resolution and fidelity required for producing fine-scale cloth wrinkles, struggle with accurately resolving detailed cloth self-collision, and, do not provide consistent draping geometries for simulations as we vary input model resolution. At the same time, consistent and meaningful quantitative metrics for evaluating the success of physics-based upsampling methods have also been missing. To address these gaps, we introduce a large-scale dataset specifically designed for cloth-drape upsampling, built with the recently developed "Progressive Cloth Simulation" (PCS) method and a new set of carefully constructed

benchmark evaluation metrics. PCS enables us to generate a dataset of multi-resolution tuples of corresponding cloth drapes, with drape consistency across resolution levels, over a diverse range of real-world cloth material parameters. Geometries at all resolutions are robustly interpenetration-free (a critical and necessary feature for high-quality cloth modeling), with increasingly finer details ending in the highest-resolution models corresponding to high-fidelity, completely unconstrained and fully converged cloth simulation output. Our dataset spans a wide range of diverse cloth configurations by collating over one million total simulated meshes constructed via careful parameterization across important input drape configuration variations. We provide geometric analyses of our dataset and benchmark five existing upsampling methods for cloth upsampling under various settings. To quantify performance, we introduce a new set of geometric and physical evaluation metrics. Here, as we show in our analyses, the high-fidelity cloth draping, introduced in this dataset, immediately exposes severe limitations in existing methods which are challenged by both the complex contact behaviors and the real-world cloth material properties demonstrated. Recognizing these gaps in existing methods regarding collision objects and material properties, we further develop and benchmark a new, learning-based baseline method for comparison. Extensive experimental results demonstrate the effectiveness, as well as the important added real-world complexity of our dataset. Its self-consistent models and the intricate high-resolution cloth details provide an important yet challenging benchmark, calling on future research in specialized model designs for data-driven cloth upsampling and simulation. A subset of our dataset is available at https://cloth-drape-dataset.github.io/.[1]

# 1 INTRODUCTION

Simulating high-fidelity cloth drapes is an important problem in computer graphics. However, modeling the nonlinear wrinkling, folding and self-contact behaviors of draped cloth requires expensive simulation on high-resolution meshes.

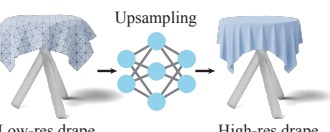

Data-driven "physics upsampling" methods offer the tantalizing promise of replacing such expensive high-resolution computations with forward evaluations of neural nets trained to predict high-resolution cloth details from low-resolution input meshes. While such a cloth upsampling method would prove immediately useful in wide-ranging applications (e.g., film (Han et al., 2021), gaming (Kavan et al., 2011), virtual reality experiences (Keckeisen et al., 2003) and virtual prototyping (Fontana et al., 2004)), existing methods do not yet achieve compelling details nor high-fidelity results. One major missing ingredient is high-quality data.

Cloth simulation as a forward process has advanced to a mature field. This should mean that generating massive amounts of cloth drape data is as easy as running an accurate cloth simulator across parameters of interest. However, data generation for physics learning tasks often quickly becomes a significant bottleneck, as cloth simulation methods able to produce high-fidelity drape results, generally remain slow and memory intensive, which, in turn, limits the size of existing cloth datasets. Moreover, prior SOTA cloth simulators (both industrial and academic) generally lack the ability to simulate the accurate draping behavior of real-world materials (largely just resolving "stretchy" materials), and generally do not include the resolution and fidelity required to capture fine cloth wrinkling in drapes (a key feature targeted by cloth upsampling) and are unable to accurately resolve detailed cloth self-collision, a key feature for detailed cloth drape folds and wrinkle formation. At the same time, all prior existing simulators are unable to provide consistent draping geometries for simulations as we vary input model resolution – a critical element for training upsampling.

In this paper, we present a large-scale dataset of cloth drapes specifically designed for cloth upsampling. To construct our dataset, we identify the recently developed "Progressive Cloth Simulation" (PCS) method (Zhang et al., 2022) as an ideal technique to generate a multi-resolution tuple of corresponding cloth drapes. Existing cloth simulators, at best, only output high-fidelity, high-resolution geometry, while the ideal data for cloth upsampling should be supervised pairs of lower-resolution cloth geometries (e.g., low-res meshes) and corresponding high-resolution, accurate cloth simulation output meshes with levels of detail. To bridge this gap, the PCS method is uniquely *consistent* across levels: avoiding artificial locking artifacts at the coarser levels while producing successively

---

[1]The full dataset will be released upon paper acceptance.

finer-detailed wrinkles that progressively appear with each level of refinement, ending in the highest-resolution models corresponding to high-fidelity, completely unconstrained and fully converged cloth simulation outputs while preserving the coarser structures from prior levels. These levels of detail are then immediately useful to supervised methods which assume pairs of low- and high-resolution data, and open the door to multi-level learning methods that could leverage this broader range of resolution. PCS has the added benefits of supporting a full range of real-world cloth material properties (which we span in our outputs), robust contact handling (ensuring non-(self-)intersecting and so high-quality geometries in all our data), and superior performance to comparable, high-fidelity cloth simulators.

Cloth behavior depends heavily on parameters across a number of axes (see Figure 1): most notably the material moduli and thickness of the cloth (e.g., leather is thick and stiff, while silk is thin and soft), the amount of self-contact (e.g., dropping a table cloth flat on the ground produces no self-folding with contacting wrinkles versus dropping it on its side), and the arrangement of resting contact with collider objects in the scene (e.g., draping cloth over the corner of a simple box shape results in vastly different wrinkles than drapes upon a complex non-convex shape). To span a diverse set of cloth configurations, we collect over one million total meshes, representing five-level PCS simulations draping cloths of wide-ranging material parameters over a variety of initial rotations and collider objects selected from the Thingi10k (Zhou & Jacobson, 2016) and PartNet (Mo et al., 2019) datasets and provide geometric analyses of the dataset, revealing its *unique cloth drape* characteristics.

Our dataset serves as a new and effective benchmark that enables the study and construction of cloth upsampling methods. With its significantly increased complexity, variation and realism in cloth drape examples, it exposes new and significant challenges for physics-based upsampling to become a practical and useful framework. At the same time, the high-quality and widely varying multi-resolution mesh tuples and our newly proposed evaluation metrics also offer opportunities to develop new, more material and configuration aware learning methods for physics upsampling and cloth simulation more broadly. To this end, we provide geometric analyses of our dataset and benchmark five existing methods for cloth upsampling under a range of settings. To evaluate cloth upsampling quality, we propose a new and detailed set of benchmark evaluation metrics that include both physical and geometric measures for the cloth upsampling task. Considering that current learning-based upsampling methods overlook collision objects and material properties in their method design, we further introduce a learning-based baseline method utilizing our benchmark and compare it with existing upsampling approaches. Extensive experimental results highlight both the efficacy and the enhanced complexity in our dataset from intricate cloth wrinkling details, marking it as an important and challenging benchmark for improving future neural upsampling and simulation methods.

In summary, we provide new, unique and valuable contributions to the learning community for learning cloth draping by offering:

1. a new, high-fidelity cloth drape dataset with significantly improved resolution, physical and geometric accuracy, and example diversity in comparison to prior cloth datasets.
2. the first multi-resolution cloth dataset with self-consistent level-of-detail geometries per drape.
3. the first large-scale (well over one million total meshes) cloth drape dataset with a wide range of cloth mesh resolutions, materials, shapes and complex and diverse collision geometries.
4. the first comprehensive benchmark suite with carefully constructed evaluation measures that consider both physical and geometric accuracy.

and additionally providing

1. a detailed geometric analysis of the collected dataset.
2. a learning-based baseline method for cloth upsampling that is collision-shape- and material-aware.
3. extensive evaluations and analyses under various settings.

## 2 RELATED WORK

**Physics-based upsampling.** Early approaches to cloth upsampling could produce fine-scale details to some extent but struggled with scalability and accurately representing contact-driven behaviors (Kavan et al., 2011; Zurdo et al., 2012). Deep neural nets have emerged as a promising tool to enhance the potential of physics-based upsampling. These methods have been particularly successful in Eulerian fluid simulation which can leverage the regularity of the fixed background grid (Chu & Thuerey, 2017; Xie et al., 2018). For cloth animation, Chen et al. (2021a) develop a deformation representation to learn a mapping from low-res garment meshes to high-res ones. In cloth simulation, recent efforts

apply hierarchical deep nets (Oh et al., 2018; Lee et al., 2019) to upsample the coarse simulation output to a fine-resolution one for cloth geometry enhancement. We include two representative methods (i.e., Kavan et al. (2011) and Lee et al. (2019)) in our benchmark for evaluation. Additionally, the tension field theory can be employed to enhance coarse geometry by introducing the amplitude and phase of intricate fine-scale wrinkling (Chen et al., 2021b). We also include the method by Chen et al. (2021b) in the benchmark evaluation.

**Geometric upsampling.**  Mesh subdivision is a technique for coarse mesh refinement, where subdivision surfaces are defined by deterministic, recursive upsampling of a discrete surface mesh (Loop, 1987; Catmull & Clark, 1978; Zorin et al., 1996). In the context of cloth upsampling, these methods can be viewed as applying a predetermined smoothness prior, unaware of external collision geometries or physical properties that might be derived or inferred from prior knowledge or training data specific to an application domain. Deep neural nets have been used to enhance the range of geometric details producible using a subdivision framework (Liu et al., 2020; 2023), which also spans the broader domain of mesh generation, compression and synthesis (Hanocka et al., 2020; Chen et al., 2023; Hertz et al., 2020). Although not explicitly designed for cloth, we evaluate the capability of two geometric methods (i.e., Loop (1987) and Neural Subdivision (Liu et al., 2020)) to capture the distinct details inherent to cloth-drape geometries.

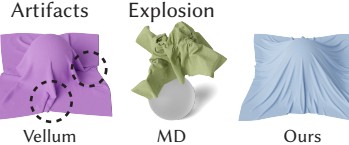

**Cloth datasets.**  Deep learning-based techniques have extensively investigated alternatives to mimic, emulate and even forward-step physically-based simulation results. Many existing approaches for neural garment deformations rely on supervised learning (Bertiche et al., 2021; Gundogdu et al., 2019; Patel et al., 2020; Tiwari et al., 2020; Wang et al., 2019; Zhang et al., 2021; Shao et al., 2023), for which it is necessary to run and collect large samples of offline physics-based simulations to gather the data required for training. In the absence of a shared, large-scale high-quality cloth draping dataset, researchers end up repeatedly generating data using accessible cloth simulators like PhysBAM (Wu et al., 2020) and Marvelous Designer (Designer, 2023), leading to datasets of limited size and quality. Concretely, corresponding high-resolution cloth simulations generated in such existing cloth simulators/software, e.g., Houdini Vellum (SideFX, 2022), Marvelous Designer (Designer, 2023), CLO (CLO, 2023), ArcSim (Narain et al., 2014), and PhysBAM (Wu et al., 2020), generate numerous errors and inaccuracies including self-intersections, sharp edge-creasing, numerical material softening, jitters, and instabilities. At the same time, these methods are unable to generate consistent geometries by simulation across varying resolutions like PCS (Zhang et al., 2022), thus prohibiting the generation of effective training sets. While a dataset of real-world captured cloth drapes would be another interesting and potentially impactful dataset, it is not able to scale nor would such a dataset be directly suitable for the cloth upsampling task, due to the noise in reconstructing meshes with different levels of detail and establishing correspondences across different resolutions while ensuring mesh quality and wrinkle consistency.

Please refer to Appendix B for a comprehensive discussion on multi-resolution simulation and physics-based enrichment and refinement.

## 3 BACKGROUND: PROGRESSIVE CLOTH SIMULATION

Throughout our dataset, we offer, per simulated drape, a multi-resolution tuple with a hierarchical structure of increasingly finer-resolution triangle meshes. The meshes in this hierarchy are organized in ascending resolution order by subscript $l \in [0, L]$, where $x_l$ and $\bar{x}_l \in \mathbb{R}^{3n_l}$ respectively represent the $n_l$ deformed and resting positions of a draped cloth's mesh nodes at level $l$. The coarsest mesh's deformed positions are saved in $x_0$, and $x_L$ holds the highest-resolution positions of the final, high-quality simulation output.

The stable *equilibria* of frictionally contacting cloth drapes are computed by equipping each finest-level mesh with shell ($\Psi$), contact barrier ($B$), friction ($D$) and, when needed, strain-limiting ($S$) potential energies. PCS then forward-steps from each initial, input, non-equilibrium configuration, to a final stable equilibrium drape. At equilibrium, PCS generates the hierarchy of final, (intersection-free and strain-limit compliant) consistent meshes $x_l$, with the finest-level mesh accurately satisfying high-accuracy equilibrium with $\|\nabla\Psi(x_L) + \nabla B(x_L) + \nabla D(x_L) + \nabla S(x_L)\| \leq \epsilon$. See Appendix A for details on the cloth model we utilize, its parameterization, and the process for PCS simulation.

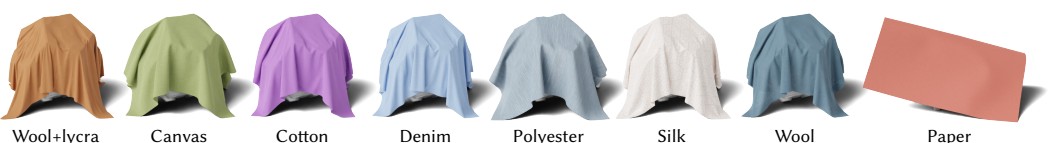

| Wool+lycra | Canvas | Cotton | Denim | Polyester | Silk | Wool | Paper |

Figure 2: **Material variations.** We select eight representative real-world materials with varying material behaviors where the material parameters can be directly plugged into the PCS solver for generating our multiresolution results. We showcase the finest level converged results here.

## 4 CLOTH DRAPE DATASET

Our dataset provides multi-resolution results for a diverse set of cloth-drape simulations parameterized over a wide range of cloth 1) shapes, 2) materials, 3) initial drape configurations (rotations), and 4) physical sizes and thicknesses; as well as varying 5) collider geometries and their 6) physical sizes.

**Cloth shapes and triangulations.** To cover both structured and unstructured triangulations, we select two shapes for the base rest shape of the cloth: a square and a circle. The square cloth employs a regular structured triangulation for its base hierarchy, whereas the circle cloth begins with an unstructured, irregular triangulation. Successive subdivision-based refinement then defines the remaining mesh resolutions for each tuple. See inset for illustrations and Table 7 for our cloth-mesh statistics.

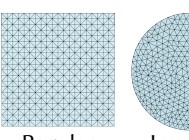

Regular   Irregular

**Cloth materials and thicknesses.** Two of the most notable factors influencing cloth draping behavior are the physical material (moduli) and the thickness of the cloth. To capture a broad spectrum of cloth behaviors including fine and coarse wrinkling characteristics, we select 8 real-world cloth materials, i.e., cotton, wool, wool + lycra, polyester, paper, canvas, silk and denim (see Figure 2). For each such material, we then generate two additional *synthetic* variations by further modifying the cloth materials's thickness. See Appendix C for the complete set of material parameters.

**Cloth initial rotations.** Varying the initial orientation of the cloth before it is dropped onto its collider can result in various final draping outcomes. We define five different initial rotations, i.e., by rotating the cloth about the x-axis by 0, 45 and 90 degrees and the z-axis by 0, 45 and 90 degrees (see inset).

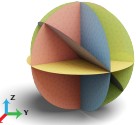

**Cloth sizes.** In contrast to purely geometric tasks, the physical size of an object (scale) significantly affects simulation results. For the square cloth, we define three sizes: 1 meter × 1 meter, 0.75 meters × 0.75 meters and 0.5 meters × 0.5 meters. For the circle cloth, we define three sizes: 1-meter diameter, 0.75-meter diameter and 0.5-meter diameter. See inset below.

**Collider objects.** To provide a varied collection of collision geometries for draping, we source 48 meshes from the Thingi10K (Zhou & Jacobson, 2016) dataset, which comprises objects with widely varying shapes and intricate details to introduce different contact and folding behaviors. Additionally, we collect 52 meshes from PartNet (Mo et al., 2019), which includes common, everyday objects.

**Collider-object sizes.** For every mesh from Thingi10K, we adjust its size by normalizing the diagonal length of its bounding box to a random value between 30 cm and 70 cm. For meshes from PartNet, we resize each mesh to match its actual dimensions listed on the IKEA website. Additionally, for each collider object, we create two more variations by resizing them to 50% and 75% of their original size. For the Thingi10K meshes, we utilize the watertight meshes preprocessed by Hu et al. (2018).

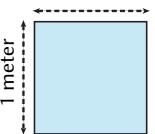 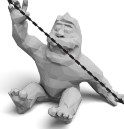

1x / 0.75x / 0.5x   0.7x / 0.3x

**Collider-object preprocessing.** Higher mesh resolutions for collider-object geometries generally result in slower simulation speeds. To enhance data-generation efficiency, we apply QSlim (Garland & Heckbert, 1997) to simplify our initial collision object meshes, ensuring the preservation of geometric features with lower-resolution final meshes. Please refer to Appendix D for preprocessing details.

### 4.1 CLOTH DRAPING SIMULATION

Figure 3 presents our data generation pipeline enabled by PCS (Zhang et al., 2022). Given a feasible initial setup, e.g., a piece of deformable cloth over a fixed collider object that is intersection-free, we generate our simulation samples across a combination of different cloth shapes, cloth materials, cloth initial rotations, cloth sizes, cloth thicknesses, colliders and collider sizes and then run PCS

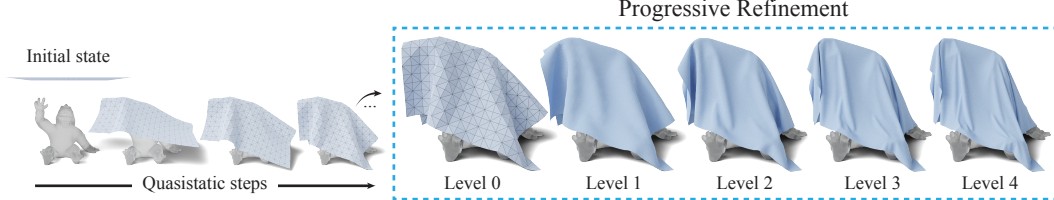

Figure 3: **Cloth simulation.** PCS allows us to progressively simulate cloth drapes from low to high resolutions with consistent results.

to produce cloth drapes at five increasingly higher resolution levels with consistent and progressive improvement over wrinkle and fold details. Table 9 reports an analysis of average runtime.

**Implementation details.**    We implement our cloth simulation pipeline in C++. We use LIBIGL (Jacobson et al., 2018) for common geometry processing subroutines. We refer readers to PCS (Zhang et al., 2022) for its own implementation details. We run simulation jobs on a dedicated CPU cluster with 2.5GHz Intel(R) Xeon(R) processors. We use 25 CPU cores, each with 386GB RAM, to parallelize the data generation jobs. Generating the entire dataset takes approximately 45 days.

### 4.2   DATASET ANALYSIS

Table 8 reports the statistics for a variety of geometric characteristics in our dataset. Specifically, we adopt the correspondence Euclidean distance $d_{\text{corr}}$, the mean curvature by quadric fitting $d_{\text{H}}^{\text{fit}}$ (Panozzo et al., 2010), the discrete differential geometry curvature using dihedral angles around a vertex $d_{\text{H}}^{\text{ddg}}$ (Knöppel et al., 2013), the Gaussian curvature $d_{\text{K}}^{\text{fit}}$ and the discrete differential geometry curvature using angle defect $d_{\text{K}}^{\text{ddg}}$ (Meyer et al., 2003). Using different metrics, we measure the similarity/consistency between the cloth drape at each coarse level (i.e., levels 0, 1, 2 and 3) and at the finest level (i.e., level 4). Please refer to Appendix E for the detailed definition of each metric.

### 4.3   DATA ACCESS AND STORAGE

Our dataset contains over one million cloth drapes with cloth drapes stored in a standard geometric file format with `.obj`. The entire dataset occupies over 3.8TB (before zipping).

We develop a website to host our dataset and facilitate the interactive exploring of cloth drapes (see Figure 6). Our prototype at `https://cloth-drape-dataset.github.io/` contains a user-friendly model viewer and instructions for accessing our dataset (upon paper acceptance).

### 4.4   LICENSE

We collect our collider objects following the licenses specified in each source dataset: the MIT license in the PartNet (Mo et al., 2019) dataset and a variety of open-source licenses in the Thingi10K (Zhou & Jacobson, 2016) dataset (see Figure 12 in Zhou & Jacobson (2016)).

## 5   CASE STUDY APPLICATION: CLOTH DRAPE UPSAMPLING

Our multi-resolution cloth drape dataset serves as a benchmark for evaluating existing cloth upsampling techniques and inspires further research to address the shortcomings of these existing methods. In this section, we consider cloth drape upsampling as a case-study application.

**Problem formulation.**    The task of cloth drape upsampling is often framed as a mesh subdivision problem. Given the cloth mesh at resolution level $i$, i.e., $M_i = (V_i, F_i)$, where $V_i$ represents the vertices and $F_i$ represents the faces, we first modify the mesh topology based on specific rules (e.g., midpoint subdivision as used by Loop (1987)) and then adjust the positions of the mesh vertices. The primary objective of the cloth upsampling task is to directly enhance cheap-to-generate, low-resolution, coarse-cloth drapes with rich, physically realistic cloth-drape details when iteratively upsampling a mesh from resolution level $i$ to level $i + 1$ (and ultimately the finest level $L$).

**Evaluation metrics.**    We propose a new set of geometric and physical measures to evaluate the quality of different upsampled results. For geometric metrics, we measure similarity by computing the correspondence Euclidean distance $d_{\text{corr}}$, the mean curvature by quadric fitting $d_{\text{H}}^{\text{fit}}$ (Panozzo et al., 2010), the Gaussian curvature $d_{\text{K}}^{\text{fit}}$, and the number of intersections #I (both cloth self-intersections and intersections with colliders). The distance and curvature metrics are computed between the

Table 1: Experimental results of cloth drape upsampling.

| Method | $d_{\text{corr}}$ | $d_{\text{H}}^{\text{fit}}$ | $d_{\text{K}}^{\text{fit}}$ | #I | $\|\nabla W\|$ | $\|\nabla \tilde{W}\|$ |
|---|---|---|---|---|---|---|
| | $\times 10^{-3}$ | $\times 10^{2}$ | $\times 10^{6}$ | $\times 10^{1}$ | $\times 10^{0}$ | $\times 10^{-1}$ |
| Kavan et al. (up to level 2) | 13.48 | 16.10 | 3.80 | 60.33 | 1.31 | 0.43 |
| TFW | 30.68 | 18.81 | 4.09 | 106.00 | 12.07 | 19.81 |
| Loop | 25.56 | 15.11 | 3.48 | 12.30 | 6.05 | 3.21 |
| U-DNN | 10.74 | 15.04 | 3.52 | 75.17 | 4.11 | 4.07 |
| Neural Subdiv | 12.35 | 15.29 | 3.54 | 26.96 | 4.72 | 1.93 |
| Ours | 7.52 | 9.62 | 3.43 | 11.57 | 3.02 | 1.75 |

predicted upsampled cloth at each resolution level and the ground truth cloth at the highest resolution level (i.e., level 4). For cloth simulation, geometric measures do not cover physical accuracy. To capture physical accuracy measurements, we also evaluate how close the resulting upsampled cloth geometry is to an accurate equilibrium configuration given by force balance as measured by the upsampled mesh's gradient norm $\|\nabla W\| = \|\nabla\Psi(x_L) + \nabla B(x_L) + \nabla D(x_L) + \nabla S(x_L)\|$. We also quantify this error from equilibrium, rescaled by computing the scaled Newton decrement norm $\|\nabla\tilde{W}\|$ from Li et al. (2021). Please refer to Appendix E for the details of these evaluation metrics.

**Baselines.** We benchmark two geometric upsampling methods: Loop (1987) and Neural Subdivision (Liu et al., 2020) and three cloth-physics-inspired upsampling methods: Kavan et al. (2011), TFW (Chen et al., 2021b) and U-DNN (Lee et al., 2019). See Appendix G for method details.

**Our method.** Observing that baseline methods overlook both the geometry of collision objects (as discussed a key source of cloth drape fold and wrinkle variation via contact) and the material moduli, we introduce a new learning-based baseline method that takes both collider and material variation into account, and compare it with the above upsampling approaches.

In this approach, the cloth material parameters are given as a vector $v$, and the collider by a point cloud $P$. We aim to train a neural net that learns to upsample the cloth drape from resolution level $i$ to $i+1$ by subdividing each triangle into four faces and then predicting the displacements for the vertices of the subdivided mesh. Our method consists of two modules: a point cloud encoder $E$ and a cloth upsampler $U$. The point cloud encoder $E$ encodes the collision object point cloud $P$ to a latent vector $f = E(P) \in \mathbb{R}^d$, where $d$ is the

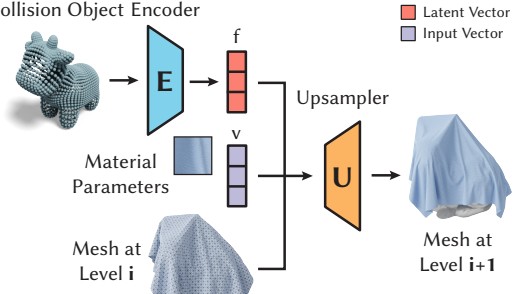

dimension of the latent vector $f$. Given a cloth drape at resolution level $i$, we first subdivide the mesh using midpoint upsampling. Then the cloth upsampler $U$ takes as input the subdivided cloth drape, the cloth material parameter vector $v$ and the latent vector $f$ of the collider, and predicts the displacements $\Delta\hat{V}_i$ for the vertices of the subdivided mesh. The predicted vertices of the mesh at the next resolution level (i.e., level $i+1$) are defined as $\tilde{V}_{i+1} = \hat{V}_i + \Delta\hat{V}_i$, where $\tilde{V}_{i+1}$ is the vertices of mesh $M_{i+1}$ and $\hat{V}_i$ is the vertices of the midpoint-subdivided mesh of $M_i$.

To train our network, we define the training loss $\mathcal{L}$ as:

$$\mathcal{L} = \sum_{i=1}^{L} \frac{1}{|V_i|} \|V_i - \tilde{V}_i\|_2, \tag{1}$$

where $V_i$ and $\tilde{V}_i$ are the ground truth and predicted vertices at resolution level $i$, respectively. Please refer to Appendix F for the details of the network and its implementation.

## 6 CASE STUDY EVALUATION

### 6.1 TASK PERFORMANCE

In this evaluation, we aim to analyze how well existing baseline upsampling methods perform, compared to our proposed material- and collider-aware method on the cloth drape upsampling task. Specifically, given a coarse cloth drape at level 0, each method upsamples it to subsequent levels, i.e.,

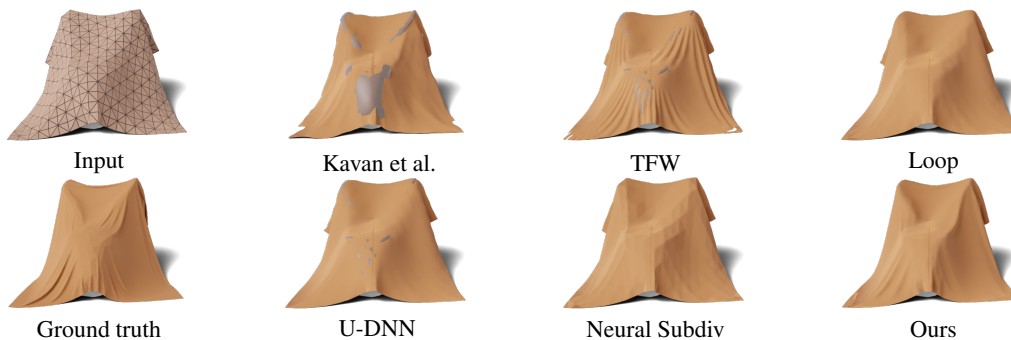

| Input | Kavan et al. | TFW | Loop |
|---|---|---|---|
| Ground truth | U-DNN | Neural Subdiv | Ours |

Figure 4: Visual comparison to existing upsampling methods.

Table 2: Generalization to unseen cloth materials.

| Method | $d_{\mathrm{corr}}$ | $d_{\mathrm{H}}^{\mathrm{fit}}$ | $d_{\mathrm{K}}^{\mathrm{fit}}$ | #I |
|---|---|---|---|---|
| | $\times 10^{-3}$ | $\times 10^2$ | $\times 10^6$ | $\times 10^1$ |
| U-DNN | 21.85 | 14.77 | 6.03 | 73.13 |
| Neural Subdiv | 22.01 | 15.47 | 6.16 | 67.47 |
| Ours | 18.04 | 11.03 | 5.42 | 51.07 |

Table 3: Generalization to unseen collider objects.

| Method | $d_{\mathrm{corr}}$ | $d_{\mathrm{H}}^{\mathrm{fit}}$ | $d_{\mathrm{K}}^{\mathrm{fit}}$ | #I |
|---|---|---|---|---|
| | $\times 10^{-3}$ | $\times 10^2$ | $\times 10^6$ | $\times 10^1$ |
| U-DNN | 19.45 | 23.56 | 10.43 | 75.11 |
| Neural Subdiv | 21.00 | 23.78 | 12.40 | 78.46 |
| Ours | 16.89 | 20.73 | 9.97 | 63.31 |

levels 1 to 4. We observe that the method by Kavan et al. (2011) is only able to produce results in a reasonable computing time up to level 2 due to the eigendecomposition step which does not scale. Table 1 lists the quantitative results evaluated at the highest resolution level (i.e., level 2 for Kavan et al. (2011) and level 4 for the other methods). Table 10 reports the quantitative results evaluated at all resolution levels. Figure 4 shows a visual comparison and more in Figures 7 and 8.

As shown in Figure 4, geometric upsampling methods (i.e., Loop and Neural Subdivision) and U-DNN fail to predict wrinkling details. While the method by Kavan et al. (2011) and TFW are both able to add wrinkles (with TFW being able to add more wrinkling details), their predictions have lots of cloth-collider intersections. These methods are not aware of the cloth material and the collider shape the cloth is draped on, suggesting that upsampling with cloth geometry alone is insufficient for the cloth upsampling task. Our method is collider- and material-aware and is able to predict plausible upsampled cloth drapes with intricate wrinkling details and fewer intersections (shown in #I).

As for the multi-level results in Table 10, we observe that as the mesh resolution level increases the results measured by the distance and curvature metrics improve for all methods but at the expense of introducing more intersections. Our results suggest that specialized designs for resolving the cloth-collider intersections are required for future cloth upsampling methods.

## 6.2 GENERALIZATION TO UNSEEN CLOTH MATERIALS AND COLLISION OBJECTS

**Unseen cloth materials.** To test the generalization to unseen cloth materials, we test on the denim material and train on the rest materials. For unseen cloth materials, we set the cloth material parameter vector $v$ to zero for our method. Table 2 reports the quantitative results. While the performance slightly degrades, our method still performs favorably against existing upsampling methods.

**Unseen collision objects.** To test the generalization to unseen colliders, we test on 10 randomly selected colliders and train on the remaining ones. Table 3 lists the quantitative results. With unseen colliders, the performance degrades for all methods, while our method still achieves the best results.

## 6.3 ABLATION STUDY ON METHOD DESIGN AND DATASET SIZE

**Method design.** To evaluate our design choices, we conduct an ablation study by removing one input condition at a time. Specifically, we compare our method with two variant methods: 1) the network is not collider-aware (denoted as *Ours w/o collider*) and 2) the network is not material-aware (denoted as *Ours w/o material*). Table 4 and Figure 5 present the quantitative and qualitative results. We observe that without the collider information, the predicted upsampled cloth has lots of intersections with the collider. Without the cloth material information, though the prediction has

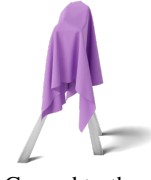
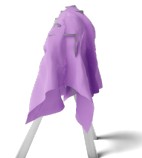
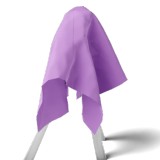
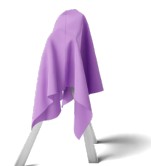

| Ground truth | Ours w/o collider | Ours w/o material | Ours |

Figure 5: Visual results of ablation study on model design.

Table 4: Ablation study on method design.

| Method | $d_{\text{corr}}$ | $d_H^{\text{fit}}$ | $d_K^{\text{fit}}$ | #I |
|---|---|---|---|---|
| | $\times 10^{-3}$ | $\times 10^2$ | $\times 10^6$ | $\times 10^1$ |
| Ours | 7.52 | 9.62 | 3.43 | 11.57 |
| Ours w/o collider | 13.15 | 16.16 | 6.51 | 27.04 |
| Ours w/o material | 14.03 | 20.94 | 8.50 | 22.13 |

Table 5: Ablation study on dataset size.

| Method | $d_{\text{corr}}$ | $d_H^{\text{fit}}$ | $d_K^{\text{fit}}$ | #I |
|---|---|---|---|---|
| | $\times 10^{-3}$ | $\times 10^2$ | $\times 10^6$ | $\times 10^1$ |
| Ours w/ 30% data | 27.81 | 20.41 | 12.47 | 81.13 |
| Ours w/ 60% data | 17.03 | 14.25 | 6.17 | 43.07 |
| Ours w/ 100% data | 7.52 | 9.62 | 3.43 | 11.57 |

fewer cloth-collider intersections, the method fails to add wrinkles to the upsampled cloth. The results suggest that both the collider and the cloth material are important for the cloth upsampling task.

**Dataset size.** Does training on more data result in better performance? To answer this question, we conduct an experiment by using 30%, 60% and 100% of the training set to train our method, respectively, and evaluate the trained model on the test set. Table 5 reports the quantitative results. We observe that the prediction quality improves as the training set size increases, showing the need for a large-scale dataset for data-driven cloth upsampling methods.

### 6.4 FAILURE CASE ANALYSIS

Figure 9 shows two failure cases of our method. In the first row, when the cloth material is too stiff (e.g., paper), our method is not able to further deform the cloth drape and tends to maintain the input cloth geometry. In the second row, when the cloth material is too soft (e.g., wool + lycra), our method is unable to predict intricate fine wrinkling details.

## 7 CONCLUSION, LIMITATIONS & FUTURE WORK

We present a large-scale dataset for physics-based cloth drape upsampling. Our work addresses the limitations of existing datasets by introducing a comprehensive collection of cloth drapes that exhibit self-consistency across multiple resolutions. We propose a new set of evaluation metrics, benchmark a number of existing geometric and physics-inspired cloth upsampling methods on our dataset and further develop a learning-based method for comparison. Experimental findings show that our technique consistently excels over other methods across various settings. Additionally, solely relying on cloth geometry for upsampling proves inadequate; information from the collider shape geometry and cloth material also plays an essential role. We further show that the quality of prediction improves as the training set size increases.

Future endeavors to expand this dataset by including shell geometries, additional collision shapes, cloth shapes and cloth initializations, garments close to the human body, and data that reflects the dynamic nature of cloth in real-world scenarios will enhance the dataset's utility. Although we have meticulously chosen a diverse set of twenty-four cloth materials for the dataset, exploring an even wider array of materials would be intriguing.

Our dataset opens new horizons for the evolution of future physics-based upsampling techniques. The preliminary success of our baseline method points to the potential of developing new upsampling methods to manage cloth-collider intersections and produce contact-safe and intersection-free results with more realistic detailed wrinkles by incorporating, for example, physics-informed GNNs (Pfaff et al., 2020) or collision losses (Santesteban et al., 2022) to resolve cloth-collider collisions. Beyond the benchmark's primary goal of advancing the cloth upsampling task across a broad spectrum of configurations and real-world materials, we believe it has potential applicability to various other cloth-centric tasks. These include fabric material parameterization, retrieval, simulation rigging, and forward drape synthesis, among others. We hope that by releasing our dataset, various communities will engage in discussions and further development surrounding this practical yet challenging physics-based cloth drape upsampling task.

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

APPENDIX

# A   BACKGROUND: PROGRESSIVE CLOTH SIMULATION

Zhang et al. (2022) have constructed the hierarchical simulation framework, progressive cloth simulation (PCS), which allows for rapid visualizations and high-quality solutions for shell quasistatic simulations on triangle-meshed geometries. Progressive simulation offers consistent and improving previews of solutions using a layered approach of increasingly higher-resolution meshes. The final, high-quality simulation output is achieved on a high-resolution mesh.

The hierarchical structure of PCS consists of a set of triangle meshes along with a matching set of prolongation operators, each corresponding to the next finer-resolution mesh. The meshes in this hierarchy are organized in ascending resolution order by subscript $l \in [0, L]$, where $\bar{x}_l$ and $x_l \in \mathbb{R}^{3n_l}$ respectively represent the $n_l$ resting and deformed positions of mesh nodes in $\mathbb{R}^3$ at level $l$. The coarsest mesh's deformed positions are saved in $x_0$, and $x_L$ holds the highest-resolution positions of the final, high-quality simulation output. Each level $l$ is linked with a prolongation operator, $P_{l+1}^l$ that maps nodal positions from the current level to the next, higher-resolution level $l+1$. For the sake of clarity, we will refer to the highest-level resolution quantities without decoration, e.g., $x = x_L$, $\bar{x} = \bar{x}_L$, and $n = n_L$.

The PCS framework is founded on the barrier-based Codimensional Incremental Potential Contact (C-IPC) (Li et al., 2021) cloth model, ensuring non-intersection and, when needed, meeting strain-limit requirements. Each simulation mesh in the PCS framework is equipped with shell ($\Psi$), contact barrier ($B$), friction ($D$), and when necessary, strain-limiting potential energies ($S$). These energies are used to calculate the stable *equilibria* of frictionally contacting shells given imposed boundary conditions and external forces. These equilibria are the local (constrained) minimizers of the total potential energy, $E_l(x) = E_l(x, \bar{x}, u)$, constructed from the sum of the aforementioned potentials, $E_l = \Psi_l + B_l + D_l + S_l$, where $u$ collects current material and boundary-condition parameters.

Progressive simulation starts with a non-equilibrium configuration, $x^t$, and material and scene parameters, $u$, and then computes stable equilibria by time-stepping its gradient flow with implicit Euler. At the highest level, this involves calculating a sequence of forward quasistatic positional updates through artificial 'time steps' (of size $h$) from $t$ to $t+1$ by minimizing an updated *incremental* potential,

$$x^{t+1} = \operatorname*{argmin}_x \frac{1}{2h^2} \|x - x^t\|_M^2 + E(x, \bar{x}, u^{t+1}), \tag{2}$$

with a mass matrix $M$, until equilibrium is reached, indicated by $\|\nabla E(x^*)\| \leq \epsilon$.

Progressive simulation employs a one-way, nonlinear multiresolution simulation-solver (Zhang et al., 2022) to evade costly direct simulation on the target high-resolution mesh. This solver comprises two phases:

**Preview, $x_l^t \to x_l^{t+1}$:** The solution at level $l$ advances quasistatically from time step $t$ to $t+1$, under possibly changing $u$, offering consistent solution previews at each level; and

**Refinement, $x_l^{t+1} \to x_{l+1}^{t+1}$:** The solution progresses spatially from level $l$ to $l+1$, enhancing the resolution for a fixed set of conditions $u^t$.

## A.1   COARSE-LEVEL PROXY ENERGIES

At every reduced-level $l < L$, previews (over varying parameters $u$) and progressively refined solutions are harmonized by solving each time step's quasistatics with a *proxy* for the highest-level potential energy,

$$F_l(x_l) = \underbrace{B_l(x_l) + D_l(x_l) + S_l(x_l)}_{C_l(x_l)} + \Psi_l\big(P^l(x_l)\big). \tag{3}$$

Here, shell elastics, $\Psi_l$, are evaluated at the highest-resolution model via a direct prolongation, $P^l(x_l)$, from level $l$ to the highest scale. At the same time, the coarse, barrier-based potential terms in $C_l(x_l)$ ensure contact and strain-limit feasibility on the current level-$l$ geometry. Each reduced level is then

processed by stepping through

$$x_l^{t+1} = \underset{x_l}{\arg\min} \frac{1}{2h^2}\|x_l - x_l^t\|_{M_l}^2 + F_l(x_l). \tag{4}$$

## A.2 Refinement and Safe Initialization

When the preview solution at level $l < L$ is confirmed, refinement to the next level $l + 1$ necessitates prolonging the current solution to this finer-resolution mesh, $x_{l+1}^t = P_{l+1}^l(x_l^*)$, after which the solver keeps moving forward by quasistatic stepping with solutions of Equation (4). However, while this prolonged geometry serves as a natural starting point for the next level's solve, prolongation overlooks contact constraints and strain limits, leading to both intersections and excessive stretching. Therefore, for safe initialization, PCS finds a safe and simple, intersection-free initializer, $x_{l+1}^s$, via in-plane upsampling. Following it, a linear search along $d = P_{l+1}^l(x_l) - x_{l+1}^s$ is conducted to find a close-by feasible point to the prolonged solution. By introducing an intersection-free point, any remaining violations of the strain limit are subsequently addressed through a stretch-reducing optimization process that brings the principal stretches below their respective limits.

## A.3 Stopping Criteria

For each coarser-level preview solve ($l < L$), the solution is determined by checking if an equilibrium is reached in terms of minimizing the proxy energy $F_l$, i.e., $\|\nabla F_l(x_l^*)\| \leq \epsilon$. Later, when reaching the finest level $L$, after refinement, the solver is, by construction, directly minimizing the target finest-level total energy, $E$, via the quasistatic stepping of the fine-level incremental potential Equation (2). This ensures that the finest-level drape geometries are always fully converged, and so high-quality equilibrium solutions. Intermediate preview geometries are then approximated solutions estimating this solution at coarser levels.

## B Extended Related Work

**Multi-resolution cloth simulation.** Our dataset is generated with the recent Progressive Cloth Simulation (PCS) method (Zhang et al., 2022). We delegate full consideration of the state-of-the-art cloth simulation in computer graphics to that work and limit our discussion to alternative choices for our multi-level data generation needs. The high-performance of methods exploiting parallelization (Selle et al., 2008; Tang et al., 2013; Schmitt et al., 2013; Tang et al., 2016; 2018; Li et al., 2020a) and/or fast but generally approximate solvers (Bouaziz et al., 2014; Zhang et al., 2019; Ly et al., 2020; Daviet, 2020; Bender et al., 2013; Müller et al., 2007; English & Bridson, 2008; Goldenthal et al., 2007) are attractive for generating massive amounts of simulation data, but are challenging to run with real-world materials, and can suffer from unacceptable artifacts or instabilities when doing so (e.g., see inset). Speed then is secondary to ensuring that the cloth simulations are both highly accurate and generated in *consistent* levels of detail. The PCS method inherits its robustness and accuracy from the IPC-model's coupling of elasticity with a mollified barrier-based frictional contact model (Li et al., 2020b), as well as even more recent improvements to include the tight strain-limit (Li et al., 2021) resolution necessary for many cloth materials. PCS generates a cascade of coarse-to-fine "preview" simulations that are guided by accurate quasistatic potentials measured on the finest level. This ensures that the coarse level results modeled by PCS are *consistent* and improve to a finest-level high-fidelity draped geometry as demonstrated by their analyses of wrinkle patterns and curvature distribution (Zhang et al., 2022).

**Physics-based enrichment and refinement.** An alternative to progressive simulation would be to run an as-accurate-as-possible simulation at each level independently. Due to both the solution errors introduced by coarser-mesh discretizations and the fundamental nonconvexity of cloth modeling, small changes in discretization generally result in large differences

[Umetani et al. 2011]

in final simulated drape geometries (e.g., different overall drape shapes and/or large changes in the constellations of contacts and wrinkles emerging in drape patterns). While fine-tuning (i.e., warm starting) finer levels with coarse-level results helps to some degree (Umetani et al., 2011), Zhang et al. (2022) demonstrate that inconsistencies and significant numerical artifacts persist across resolution levels. Another alternative to progressive simulation would be to run a full simulation at the coarsest

Table 6: Material parameters.

| Material | Thickness $\times10^{-4}$ | Density $\times10^{2}$ | $E_{\text{mem}}$ $\times10^{3}$ | $E_{\text{bend}}$ $\times10^{4}$ | Poisson $\times10^{-1}$ | Strain limit $\times10^{0}$ |
|---|---|---|---|---|---|---|
| Wool + lycra | 3.28 | 5.43 | 0.76 | 0.76 | 0.71 | 1.13 |
| Wool + lycra - s1 | 6.60 | 5.43 | 0.76 | 0.76 | 0.71 | 1.13 |
| Wool + lycra - s2 | 1.80 | 5.43 | 0.76 | 0.76 | 0.71 | 1.13 |
| Canvas | 5.30 | 2.94 | 20.09 | 20.09 | 3.00 | 1.11 |
| Canvas - s1 | 6.60 | 2.94 | 20.09 | 20.09 | 3.00 | 1.11 |
| Canvas - s2 | 1.80 | 2.94 | 20.09 | 20.09 | 3.00 | 1.11 |
| Cotton | 3.18 | 4.73 | 8.21 | 8.21 | 2.43 | 1.06 |
| Cotton - s1 | 6.60 | 4.73 | 8.21 | 8.21 | 2.43 | 1.06 |
| Cotton - s2 | 1.80 | 4.73 | 8.21 | 8.21 | 2.43 | 1.06 |
| Denim | 6.60 | 4.00 | 24.48 | 24.48 | 3.00 | 1.28 |
| Denim - s1 | 5.68 | 4.00 | 24.48 | 24.48 | 3.00 | 1.28 |
| Denim - s2 | 1.80 | 4.00 | 24.48 | 24.48 | 3.00 | 1.28 |
| Polyester | 2.52 | 6.53 | 4.78 | 4.78 | 3.81 | 1.06 |
| Polyester - s1 | 6.60 | 6.53 | 4.78 | 4.78 | 3.81 | 1.06 |
| Polyester - s2 | 1.80 | 6.53 | 4.78 | 4.78 | 3.81 | 1.06 |
| Silk | 1.80 | 0.83 | 5.70 | 5.70 | 3.00 | 1.41 |
| Silk - s1 | 6.60 | 0.83 | 5.70 | 5.70 | 3.00 | 1.41 |
| Silk - s2 | 5.68 | 0.83 | 5.70 | 5.70 | 3.00 | 1.41 |
| Wool | 5.68 | 4.13 | 1.70 | 1.70 | 2.77 | 1.09 |
| Wool - s1 | 6.60 | 4.13 | 1.70 | 1.70 | 2.77 | 1.09 |
| Wool - s2 | 1.80 | 4.13 | 1.70 | 1.70 | 2.77 | 1.09 |
| Paper | 3.00 | 8.00 | 30000.00 | 30000.00 | 3.00 | 1.01 |
| Paper - s1 | 6.60 | 8.00 | 30000.00 | 30000.00 | 3.00 | 1.01 |
| Paper - s2 | 1.80 | 8.00 | 30000.00 | 30000.00 | 3.00 | 1.01 |

level, and then run a constrained finer-level simulation which should *track* the coarse level to ensure consistency (Bergou et al., 2007). This could be applied repeatedly to build a multi-level hierarchy of simulations. While suitable for low-accuracy requirement scenarios, the major issue with this paradigm is that accuracy at the finest level is sacrificed to satisfy constraints pulling the finest level solution to match the least accurate (i.e., coarsest) simulation rather than providing accurate simulation detailing. A similar lack of high-resolution accuracy exists for methods that augment low-res cloth with high-res details computed via separate subspaces (Müller & Chentanez, 2010; Zurdo et al., 2012; Hahn et al., 2014; Gillette et al., 2015; Jin et al., 2020).

## C  MATERIAL PARAMETERS

Table 6 lists the material parameters considered in this paper. These parameters correspond to real-world materials which are obtained by directly referencing online sources. The synthetic materials we created have names ending with "- s1" and "- s2".

## D  DETAILS OF COLLIDER OBJECT PREPROCESSING

As stated in Section 4, we include a collider-object preprocessing step. We adopt the QSlim (Garland & Heckbert, 1997) algorithm to decimate the collider object mesh. Specifically, we decimate the collider object mesh so that the distortion error is less than 0.1% of the bounding box diagonal length while having the minimum number of faces. The distortion error is measured by sampling 1 million points on the surface of the decimated collider object mesh and computing the point-to-mesh distance of those sampled points to the original collider object mesh. This allows us to preserve the geometric features with lower-resolution final meshes.

# E   DETAILS OF EVALUATION METRICS

**Correspondence Euclidean distance $d_{\text{corr}}$.**   To quantitatively assess the similarity between the predicted fine mesh and the ground truth, we adopt the Euclidean distance metric for similarity measurement. The correspondence Euclidean distance computes the Euclidean distance between the vertices of the predicted mesh and the vertices of the ground truth mesh (i.e., the mesh at the finest level). For meshes (from different levels) with vertices fewer than the ground truth mesh vertices, we apply midpoint upsampling until the number of vertices matches the ground truth mesh vertices before computing the distance.

**Curvature metrics.**   Since meshes are surfaces over a common parameterization of a domain $\Omega$, another way to measure the difference between two surfaces is to compute the difference of curvature integrated over this parametric domain. Specifically, we compute the total integral of the squared difference in curvature as

$$d = \int_{\Omega} \|H_{\text{pred}} - H_{\text{gt}}\|^2 dA, \tag{5}$$

where $H_{\text{pred}}$ denotes the curvature of the predicted upsampled cloth and $H_{\text{gt}}$ denotes the curvature of the ground truth cloth (i.e., the cloth drape at the finest level.)

There are two common measures of surface curvature in differential geometry: mean curvature (the average of the principal curvatures at a given point on a surface) and Gaussian curvature (the product of the principal curvatures at a given point on a surface.) We adopt both of them here for completeness. In discrete differential geometry, curvature is approximated per vertex on a discrete surface. The curvature difference $d$ in Equation (5) is then computed as the sum of the squared difference in per-vertex quantities between the predicted upsampled cloth shape and the ground truth (i.e., the cloth drape at the finest level.)

One preferred way to estimate these vertex quantities is to compute the mean curvature approximated through local quadric fitting (Panozzo et al., 2010). We denote this mean curvature by quadric fitting metric as $d_{\text{H}}^{\text{fit}}$. We can similarly estimate these vertex quantities by computing the Gaussian curvature approximated by local quadric fitting and denote this metric as $d_{\text{K}}^{\text{fit}}$. In addition to fitting methods, discrete methods represent another category for estimating surface curvature, which is fundamentally based on discrete differential geometry concepts. As an emphasis of the robustness of the experiments, we also compute discrete differential geometry curvatures using dihedral angles around a vertex (Knöppel et al., 2013) and denote the metric as $d_{\text{H}}^{\text{ddg}}$, and using angle defect (Meyer et al., 2003) and denote the metric as $d_{\text{K}}^{\text{ddg}}$.

**DDG curvature using dihedral angles $d_{\text{H}}^{\text{ddg}}$.**   Curvature can be computed using dihedral angles around a vertex, i.e., discrete differential geometry curvature using dihedral angles around a vertex $d_{\text{H}}^{\text{ddg}}$. The dihedral angle is the angle between two adjacent triangles along an edge. The curvature at a vertex can be related to the sum of the dihedral angles around that vertex. This is based on the Gauss-Bonnet theorem, which states that the total curvature of a surface is related to the integral of the Gaussian curvature over the surface.

**DDG curvature using angle defect $d_{\text{K}}^{\text{ddg}}$.**   Curvature can also be computed using angle defect, i.e., discrete differential geometry curvature using angle defect $d_{\text{K}}^{\text{ddg}}$. This method is based on the notion of the angle defect associated with each vertex in a mesh. The angle defect captures the deviation of the sum of angles around a vertex from the expected total angle in a flat (Euclidean) space.

Similar to the Euclidean distance metric, for meshes (from different levels) with vertices fewer than the ground truth mesh vertices (i.e., the mesh at the finest level), we apply midpoint upsampling until the number of vertices matches the ground truth mesh vertices before computing the curvature metrics.

**Number of intersections #I.**   We compute the number of cloth self-intersections and the number of cloth-collider intersections in triangle-triangle pairs and sum them.

**Gradient $\|\nabla W\|$.**   To reflect the physical accuracy of the shapes, we also evaluate how close the resulting upsampled cloth geometry is to an accurate equilibrium configuration given by force balance as measured by the upsampled mesh's gradient norm $\|\nabla W\| = \|\nabla \Psi(x_L) + \nabla B(x_L) + \nabla D(x_L) +$

$\nabla S(x_L)\|$. The gradient norm of the upsampled mesh evaluates how closely the upsampled cloth shape aligns with an equilibrium state determined by force balance.

**Residual** $\|\nabla \tilde{W}\|$. Analogous to the gradient norm, the residual norm gauges closeness through a scaled metric by examining the Newton decrement. We quantify this error from equilibrium, rescaled by computing the scaled Newton decrement norm $\|\nabla \tilde{W}\|$ from Li et al. (2021).

# F  ADDITIONAL DETAILS OF OUR METHOD

**Network details.** We use the pre-trained PointNet Qi et al. (2017) from Mescheder et al. (2019) as the point cloud encoder $E$. Our upsampler builds on Neural Subdivision (Liu et al., 2020). To make the network collider-aware and material-aware, we modify the network's input to also accept a vector from the point cloud encoder and a vector from the material parameters. We use half-flap representations as in Neural Subdivision. Different from Neural Subdivision which defines a local frame for each half-flap, we define a single global frame and everything is defined with respect to it. This allows us to encode the cloth-collider spatial relationship which is important for the upsampler $U$. We refer the reader to Liu et al. (2020) for more details about Neural Subdivision.

The material parameter vector $v$ is a 5-dimensional vector of (thickness, density, membrane, Poisson, strain limit). See Table 6 for the material parameters. Each dimension is independently normalized to 0 and 1 before being passed to the network as input.

**Implementation details.** We implement our method in PyTorch (Paszke et al., 2019) We use the ADAM (Kingma & Ba, 2014) optimizer to train our network. The learning rate is set to $1 \times 10^{-3}$ with a learning rate decay of $1 \times 10^{-6}$. We train our network on one A100 GPU with 80GB memory.

# G  DETAILS OF EXISTING UPSAMPLING METHODS

**Loop Subdivision (Loop, 1987).** We adopt the well-established Loop subdivision and apply it to iteratively subdivide the input coarse mesh to obtain the upsampled fine mesh.

**Neural Subdivision (Liu et al., 2020).** We apply Neural Subdivision to our data to learn cloth upsampling. Neural Subdivision was designed to work with meshes without boundaries. To address this limitation, we apply zero padding to handle meshes with boundaries.

**Kavan et al. (2011).** Kavan et al. require pre-computing a pair of coarse and fine training simulations aligned with tracking constraints to avoid bifurcation and then training the upsampling operators to learn mid-scale details with a new regularization method. Our self-consistent multi-resolution data inherently fulfill these prerequisites, making it ideal for direct utilization in the training process.

**TFW (Chen et al., 2021b).** Chen et al. present a technique that effectively captures the intricate details of thin shells in high definition using coarse meshes. Their method involves an initial estimation of the shell's coarse shape using tension field theory, followed by the addition of parameterized wrinkles using amplitude and phase fields to enhance the base mesh. We adopt the second phase of the method for adding high-frequency wrinkles given our coarse meshes as inputs and include the converged results.

**U-DNN (Lee et al., 2019).** U-DNN upsamples cloth drapes by learning a mapping from the coarse cloth drapes at times $t-2$, $t-1$ and $t$ to the fine cloth drape at time $t$. In our setting, we focus on *static* cloth drape upsampling. Therefore, we modify U-DNN's network architecture to learn a mapping between the coarse cloth drape at time $t$ and the fine cloth drape at time $t$.

Table 7: **Cloth statistics.** We report the numbers of vertices #V and faces #F at each level for each cloth shape.

| Shape | Triangulation | Level 0 | | Level 1 | | Level 2 | | Level 3 | | Level 4 | |
|---|---|---|---|---|---|---|---|---|---|---|---|
| | | #V | #F | #V | #F | #V | #F | #V | #F | #V | #F |
| Square | Regular | 441 | 800 | 1,681 | 3,200 | 6,561 | 12,800 | 25,921 | 51,200 | 103,041 | 204,800 |
| Circle | Irregular | 442 | 815 | 1,698 | 3,260 | 6,655 | 13,040 | 26,349 | 52,160 | 104,857 | 208,640 |

Table 8: **Dataset analysis.** $d_{\text{corr}}$: correspondence euclidean distance. $d_{\text{H}}^{\text{fit}}$: quadric fitting mean curvatures. $d_{\text{H}}^{\text{ddg}}$: discrete differential geometry curvatures using dihedral angles around a vertex. $d_{\text{K}}^{\text{fit}}$: Gaussian curvatures. $d_{\text{K}}^{\text{ddg}}$: discrete differential geometry curvatures using angle defect. See Section 4.2 for details.

| Shape | Level | $d_{\text{corr}}$ | $d_{\text{H}}^{\text{fit}}$ | $d_{\text{H}}^{\text{ddg}}$ | $d_{\text{K}}^{\text{fit}}$ | $d_{\text{K}}^{\text{ddg}}$ | | Shape | Level | $d_{\text{corr}}$ | $d_{\text{H}}^{\text{fit}}$ | $d_{\text{H}}^{\text{ddg}}$ | $d_{\text{K}}^{\text{fit}}$ | $d_{\text{K}}^{\text{ddg}}$ |
|---|---|---|---|---|---|---|---|---|---|---|---|---|---|---|
| | | $\times 10^{-2}$ | $\times 10^{2}$ | $\times 10^{2}$ | $\times 10^{5}$ | $\times 10^{6}$ | | | | $\times 10^{-2}$ | $\times 10^{2}$ | $\times 10^{2}$ | $\times 10^{5}$ | $\times 10^{6}$ |
| | 0 | 2.25 | 6.01 | 6.56 | 8.69 | 5.24 | | | 0 | 1.63 | 5.96 | 6.90 | 9.85 | 3.97 |
| Square | 1 | 1.19 | 5.10 | 5.55 | 8.48 | 4.03 | | Circle | 1 | 0.79 | 4.91 | 5.70 | 10.1 | 2.93 |
| | 2 | 0.48 | 3.53 | 3.74 | 8.36 | 3.48 | | | 2 | 0.31 | 2.96 | 3.39 | 10.0 | 2.19 |
| | 3 | 0.16 | 1.53 | 1.46 | 6.68 | 2.88 | | | 3 | 0.09 | 0.92 | 0.89 | 6.83 | 0.96 |

Table 9: Runtime analysis.

| | Seconds |
|---|---|
| Level 0 | 16.40 |
| Level 1 | 7.96 |
| Level 2 | 6.92 |
| Level 3 | 281.61 |
| Level 4 | 18.34 |
| Others | 33.37 |

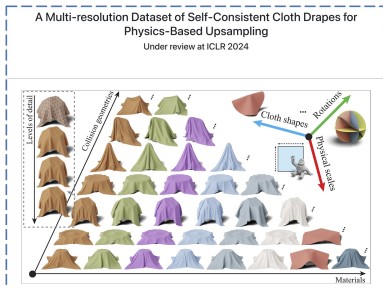
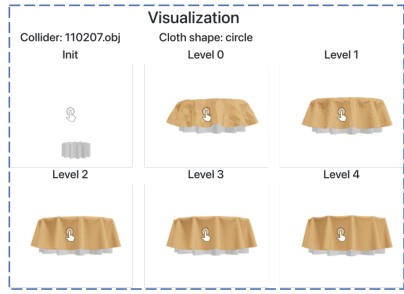

Figure 6: Dataset access.

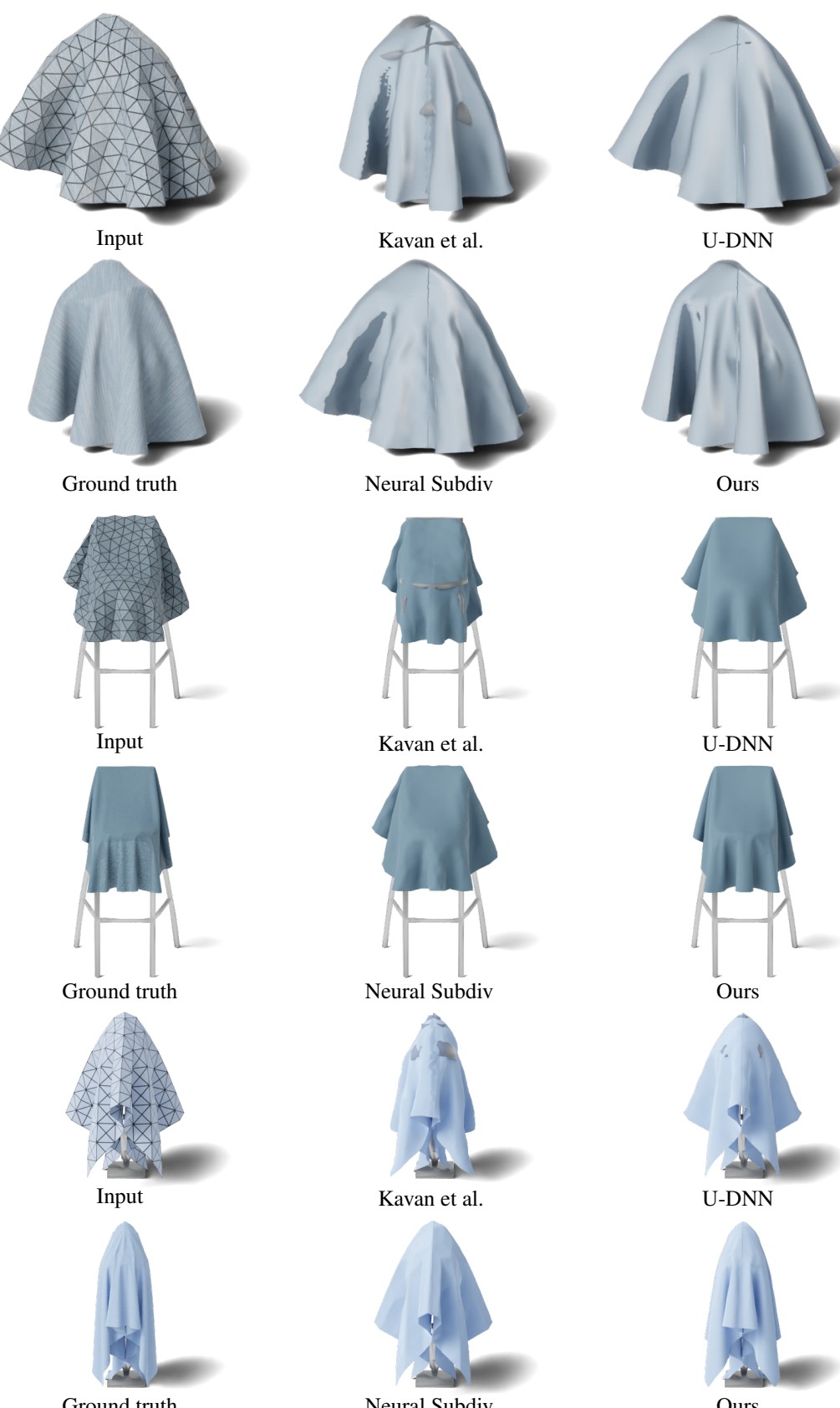

Figure 7: Visual comparison to existing upsampling methods.

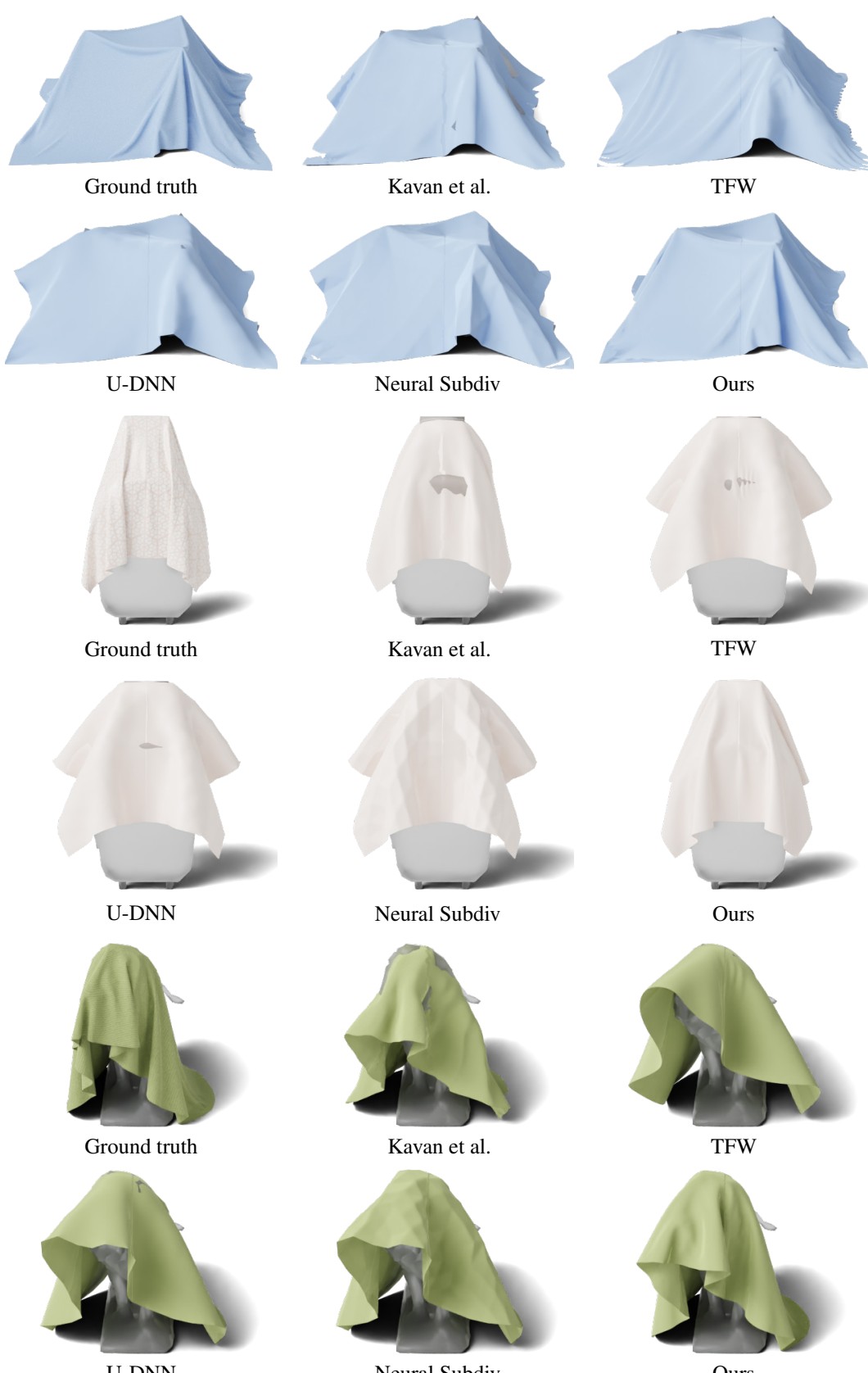

Figure 8: Visual comparison to existing upsampling methods.

Table 10: **Experimental results of cloth drape upsampling.** Each method takes as input the cloth drape mesh at level 0 and upsamples it to levels 1, 2, 3 and 4. Kavan et al. (2011) can only produce results at levels 1 and 2.

| Method | Level | $d_{\mathrm{corr}}$ | $d_{\mathrm{H}}^{\mathrm{fit}}$ | $d_{\mathrm{K}}^{\mathrm{fit}}$ | #I | $\|\nabla W\|$ | $\|\nabla \tilde{W}\|$ |
|---|---|---|---|---|---|---|---|
| | | $\times 10^{-3}$ | $\times 10^{2}$ | $\times 10^{6}$ | $\times 10^{1}$ | $\times 10^{0}$ | $\times 10^{-1}$ |
| Kavan et al. | 1 | 17.41 | 16.56 | 4.02 | 32.90 | 0.82 | 0.47 |
| | 2 | 13.48 | 16.10 | 3.80 | 60.33 | 1.31 | 0.43 |
| TFW | 1 | 217.36 | 23.03 | 6.65 | 18.60 | 1.56 | 2.67 |
| | 2 | 73.42 | 20.85 | 6.37 | 31.57 | 3.83 | 4.01 |
| | 3 | 38.85 | 20.34 | 5.03 | 57.06 | 9.21 | 11.01 |
| | 4 | 30.68 | 18.81 | 4.09 | 106.00 | 12.07 | 19.81 |
| Loop | 1 | 123.50 | 19.17 | 7.31 | 2.70 | 0.97 | 0.18 |
| | 2 | 93.51 | 15.68 | 5.31 | 6.80 | 1.17 | 0.82 |
| | 3 | 76.52 | 15.13 | 3.57 | 8.60 | 2.77 | 6.31 |
| | 4 | 25.56 | 15.11 | 3.48 | 12.30 | 6.05 | 3.21 |
| U-DNN | 1 | 22.81 | 16.68 | 4.21 | 16.40 | 0.66 | 0.60 |
| | 2 | 16.44 | 15.51 | 3.62 | 25.06 | 1.95 | 5.81 |
| | 3 | 11.43 | 15.06 | 3.56 | 42.03 | 3.47 | 8.71 |
| | 4 | 10.74 | 15.04 | 3.52 | 75.17 | 4.11 | 4.07 |
| Neural Subdiv | 1 | 26.54 | 16.81 | 4.14 | 11.33 | 0.91 | 0.42 |
| | 2 | 20.77 | 15.91 | 3.90 | 12.37 | 1.88 | 0.72 |
| | 3 | 18.43 | 15.32 | 3.71 | 17.96 | 1.06 | 2.89 |
| | 4 | 12.35 | 15.29 | 3.54 | 26.96 | 4.72 | 1.93 |
| Ours | 1 | 19.45 | 15.95 | 4.13 | 5.96 | 0.73 | 0.44 |
| | 2 | 10.57 | 12.21 | 3.84 | 6.97 | 1.44 | 0.98 |
| | 3 | 7.53 | 9.59 | 3.45 | 9.73 | 1.55 | 2.01 |
| | 4 | 7.52 | 9.62 | 3.43 | 11.57 | 3.02 | 1.75 |

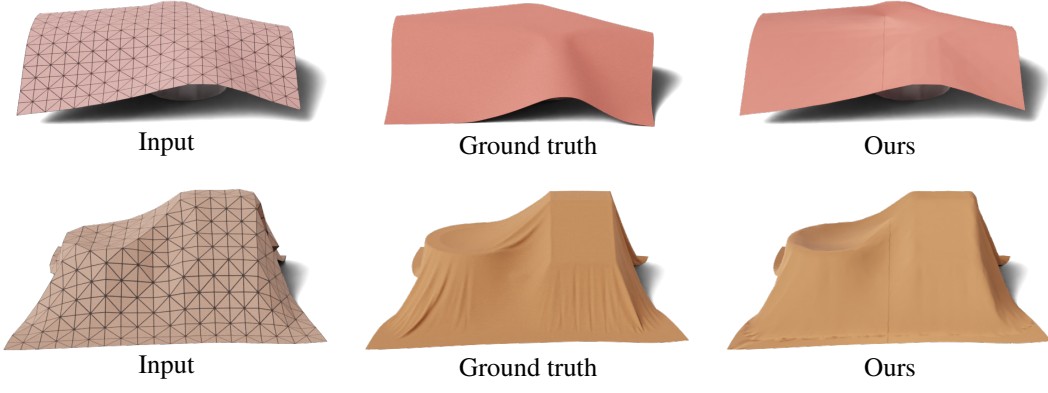

Input | Ground truth | Ours

Input | Ground truth | Ours

Figure 9: Failure cases.

