# OpenReview forum: "A  Multi-resolution Dataset of Self-consistent Cloth Drapes for Physics-based Upsampling"
_ICLR.cc/2024/Conference — Submitted to ICLR 2024_

### Official Review · Reviewer_qJGw · 2023-10-23

**Soundness:** 3 good
**Presentation:** 2 fair
**Contribution:** 1 poor
**Rating:** 5
**Confidence:** 5

**Summary:**

The research paper introduces and discusses a novel dataset of cloth simulations that interact with various collider objects. The dataset  includes a wide variety of materials, ensuring a comprehensive representation of different cloth types and their behaviors. Furthermore, the authors provide meshes at multiple resolutions, maintaining consistency across these different levels of details.

The primary objective of this dataset is to facilitate the development and evaluation of deep learning networks, specifically in the context of upscaling cloth meshes. The authors propose such deep learning pipeline and demonstrate its effectiveness over existing baselines.

**Strengths:**

The paper uses the Progressive Cloth Simulation (PCS) simulator, which ensures that the cloth meshes are consistent across different resolutions. This is a significant strength as it provides a robust foundation for establishing a training set specifically tailored for upsampling tasks, by providing correspondences between different levels of details.

The dataset covers a wide variety of cloth materials. This diversity is a major asset, as it allows for the training and testing of models in a multitude of scenarios, ensuring that the resulting models are versatile and robust.

The paper is well written.

**Weaknesses:**

A primary concern regarding this paper is its relevance in the current landscape of cloth simulation and rendering. The field has seen a significant shift towards self-supervised learning approaches, as evidenced by recent papers such as SNUG (Santesteban et al., CVPR 2022), HOOD (Grigorev et al., CVPR 2023), and DrapeNet (De Luigi et al., CVPR 2023). These works have moved away from traditional supervised learning methods, which stands in contrast to the claim made in Section 2 of the paper that “Most recent approaches for neural garment deformations rely on supervised learning”. This discrepancy raises questions about the paper's alignment with current trends and best practices in the field.

The paper introduces a dataset for training and testing deep learning models, but there are concerns regarding its generalizability. The dataset includes a limited number of collider objects (100), which may not be sufficient for ensuring that models trained on this data can generalize well to unseen geometries, as indicated by the results in Table 3. Additionally, the dataset and task are focused solely on static clothes subjected to gravity and resting on collider objects. This narrow scope may limit the utility of the dataset, as it does not reflect the dynamic nature of cloth in real-world scenarios.

The technical contributions of the paper appear to be modest. The proposed network architecture, consisting of a PointNet encoder and Neural Subdivision decoder, is relatively simple. Furthermore, the dataset is generated using a direct application of the PCS simulator, and the “new metrics” mentioned in the abstract are standard measures such as Euclidean distance and curvature. The use of different curvature computations as metrics for assessing mesh similarity is not clearly justified or explained, leaving the reader uncertain about the soundness and efficacy of these measures.


A minor point: the abstract seems too long.

**Questions:**

Why are supervised methods still relevant nowadays for cloth draping? this directly questions the relevance of a dataset for training such methods.

How and why is a curvature a measure of "similarity"? How can it be computed “between meshes” as said in the supplementary material?

The dataset is said to contain “over a million” samples: how many exactly?

---

> ### Author Response · Authors · 2023-11-17
> **Author response (1/2)**
>
> # General answer for clarifying the big picture:
>
> We value the recognition of our dataset's comprehensiveness and diversity and appreciate the acknowledgment of our objective as a dataset and benchmark paper to aid in the development and evaluation of existing upsampling methods as our contribution.
>
> However, we would like to clarify an important aspect that was overlooked in this review: **the primary application of our dataset is specifically focused on cloth detail upsampling**. This task is uniquely defined as the direct enhancement of low-resolution, coarse cloth drapes (which are usually cheap and fast to compute/generate) into high-resolution, detailed, physically realistic versions, which inherently requires a multi-scale hierarchy (as defined in the abstract and introduction and detailed in Section 5 of our paper). **It is crucial to distinguish this cloth upsampling task from neural cloth simulation, a related yet completely different application** where the aim is to train a neural network that does forward time stepping, predicting realistic cloth motion (dynamics) with regression on a single scale. **These two are fundamentally different tasks, and our dataset is tailored to the former, not the latter**. We note that all of the three referenced papers here — SNUG by Santesteban et al. (CVPR 2022), HOOD by Grigorev et al. (CVPR 2023), and DrapeNet by De Luigi et al. (CVPR 2023) — all fall into the second category, which is not the task of this benchmark.
>
> In terms of future trends, since **the task we are tackling is a fundamentally different one from learning garment motions**, it remains to be seen whether self-supervised approaches will be the dominant strategy for cloth detail upsampling. At the same time, we note that, for the state of the art, supervised learning methods, e.g., U-DNN (included in our benchmark), are currently in use. Likewise, benchmarks serve a dual purpose: they are not just tools enabling the development of new methods, but also for evaluating and comparing new and existing methods. It is commonly accepted that the results produced from current neural approaches, including those for learning dynamic garment motions still significantly lag in quality compared to SOTA high-fidelity cloth simulation outputs. Our analysis of existing upsampling methods with the high-fidelity cloth models in our PCS-generated dataset already exposes severe limitations in existing upsampling methods (as demonstrated in Table 1 and Figures 4, 7 and 8). These methods, while claiming various capabilities to generalize, when evaluated on the new dataset, demonstrate underperformance by missing the physically realistic and intricate folding behaviors formed through precise contact resolution across a wide range of real-world materials. This underperformance clearly demonstrates that the quality, complexity and diversity of the new dataset extends well beyond the original scope of the training data with which these methods were initially developed, and so demonstrates the need for both new research to bridge this gap and the utility of our current proposed benchmark.

---

> ### Author Response · Authors · 2023-11-17
> **Author response (2/2)**
>
> # Answers to specific questions:
>
> > These works have moved away from traditional supervised learning methods, which stands in contrast to the claim made in Section 2 of the paper that “Most recent approaches for neural garment deformations rely on supervised learning”. This discrepancy raises questions about the paper's alignment with current trends and best practices in the field.
>
> To clarify we should have scoped this discussion - currently most approaches for neural physics-based upsampling rely on supervised learning. It's certainly true that neural simulation (please see our discussion of neural physics upsampling vs neural forward simulation) methods more broadly apply a range of both supervised and self-supervised methods. We have adjusted our paper's discussion to reflect this.
>
> ---
>
> > The paper introduces a dataset for training and testing deep learning models, but there are concerns regarding its generalizability. The dataset includes a limited number of collider objects (100), which may not be sufficient for ensuring that models trained on this data can generalize well to unseen geometries, as indicated by the results in Table 3.
>
> In our network design, the PointNet encoder has been pre-trained on a large collection of shapes for learning occupancy fields  [Mescheder et al. 2019] (see Appendix F for details). This network design allows our network to be aware of the collider object geometry. We note that it is expected that a trained network performs better on in-distribution data (in our case, seen collider objects as in Table 1) and has performance degradation on out-of-distribution data (in our case, unseen collider objects as in Table 3). As shown in Table 3, while all methods suffer from performance drops on unseen collider objects, our learning-based baseline method still outperforms all competing methods by a large margin (at least 13% improvement on the Euclidean distance metric d_corr over U-DNN and 18% over Neural Subdivision). We look forward to further enriching our dataset with additional collision geometries, considering it an exciting avenue for future work.
>
> ---
>
> > Additionally, the dataset and task are focused solely on static clothes subjected to gravity and resting on collider objects. This narrow scope may limit the utility of the dataset, as it does not reflect the dynamic nature of cloth in real-world scenarios.
>
> We reiterate that our focus is on cloth detail upsampling, a task that is fundamentally different from learning neural dynamic garment motions.
>
> Please see more discussion on the dataset's suitability for cloth detail upsampling within the current scope here (https://openreview.net/forum?id=aAhgJ1fQ4V&noteId=rqISyiyGnu).
>
> ---
>
> > The technical contributions of the paper appear to be modest. The proposed network architecture, consisting of a PointNet encoder and Neural Subdivision decoder, is relatively simple.
>
> We note that **the newly proposed baseline method for cloth upsampling is not the central emphasis of our work** – our paper's primary contributions, as a dataset and benchmark paper (as marked in the primary area section), lie in collecting high-quality data (currently no similar public datasets exist), benchmarking existing methods for this task and developing evaluation protocols for future research. Moreover, our detailed analysis of existing upsampling methods has unveiled a crucial insight: the diversity and complexity of the cloth drapes in our dataset expose severe limitations in existing upsampling methods. To demonstrate the potential for addressing this gap, we have further developed a learning-based baseline method for cloth upsampling that improves existing upsampling methods. This addition, not claimed as a novelty by us, serves as a valuable contribution we offer. As shown in Figures 4, 7 and 8, none of the existing upsampling methods can predict reasonable results. Our simple yet effective approach of incorporating collision geometries and material properties into the neural network has yielded plausible outcomes featuring detailed wrinkling and fewer intersections. We hope this can inspire future research to dive deeper and make further advancements.
>
> ---
>
> > How and why is a curvature a measure of "similarity"? How can it be computed “between meshes” as said in the supplementary material?
>
> The details of these curvatures were well explained in the supplemental of the PCS paper. Please see section 1 here (https://pcs-sim.github.io/pcs-supplement.pdf). We are happy to include them in our appendix if considered necessary. We will release the code that computes these metrics.
>
> ---
>
> > The dataset is said to contain “over a million” samples: how many exactly?
>
> There are 100 collider shapes, 3 collider size scalings, 2 cloth shapes, 3 cloth scalings, 24 cloth materials, 5 cloth initial rotations and 5 resolutions. This amounts to 100 x 3 x 2 x 3 x 24 x 5 x 5 = 1,080,000 samples.

---

> > ### Author Response · Authors · 2023-11-18
> >
> > > How and why is a curvature a measure of "similarity"? How can it be computed “between meshes” as said in the supplementary material?
> >
> > Following above, we have included the details of the evaluation metrics and updated our paper. Please refer to appendix E for the highlighted text sections. Thanks!

---

> > > ### Author Response · Authors · 2023-11-23
> > >
> > > Dear Reviewer qJGw,
> > >
> > > As the discussion period is coming to an end, we aim to ensure that we have thoroughly addressed all your concerns. Could you kindly confirm if the clarifications provided are sufficient and clear? If there are any outstanding issues or further questions, we welcome you to share them with us.
> > >
> > > We greatly appreciate your time and input.
> > >
> > > Best regards, The Authors.

---

### Official Review · Reviewer_z6HR · 2023-10-28

**Soundness:** 3 good
**Presentation:** 3 good
**Contribution:** 2 fair
**Rating:** 6
**Confidence:** 4

**Summary:**

This paper proposes a new large-scale dataset for cloth-drape upsampling. The dataset includes multi-resolution clothes with large variations in terms of the materials, shapes, and obstacles. The dataset is also extensively anaylized and evaluated. Some applications of this dataset are cloth drape upsampling.

**Strengths:**

* High quality of clothes with multi-resolutions.
* Various materials and obstacles are included.
* Large-scale dataset of cloth draping.
* Detailed evaluations of the dataset.

**Weaknesses:**

1. Some details about the dataset are not clear. E.g., does the dataset only include static data, which is the equilibrium state? How to split the coarse mesh into more detailed one? Is it a fixed method to divide the triangle?
2. Some related work on the cloth dataset is missing: Towards Multi-Layered 3D Garments Animation, ICCV'23.

**Questions:**

Please refer to the weakness.

---

> ### Author Response · Authors · 2023-11-17
> **Author response**
>
> We thank the reviewer for providing valuable feedback.
>
> > Some details about the dataset are not clear. E.g., does the dataset only include static data, which is the equilibrium state?
>
> Yes, our dataset only includes static data of equilibrium states. Please see Section 3 for details.
>
> ---
>
> > How to split the coarse mesh into a more detailed one? Is it a fixed method to divide the triangle?
>
> As mentioned in the problem formulation paragraph in Section 5, we use midpoint subdivision to divide a triangle into four triangles.
>
> ---
>
> > Some related work on the cloth dataset is missing: Towards Multi-Layered 3D Garments Animation, ICCV'23.
>
> Thank you for pointing this paper out. We note that this paper is published at ICCV 2023, which is **after** the ICLR paper submission deadline. We have now cited this paper in the cloth dataset paragraph of Section 2.

---

> > ### Author Response · Authors · 2023-11-23
> >
> > Dear Reviewer z6HR,
> >
> > As the discussion period is coming to an end, we aim to ensure that we have thoroughly addressed all your concerns. Could you kindly confirm if the clarifications provided are sufficient and clear? If there are any outstanding issues or further questions, we welcome you to share them with us.
> >
> > We greatly appreciate your time and input.
> >
> > Best regards, The Authors.

---

### Official Review · Reviewer_QcF6 · 2023-10-29

**Soundness:** 3 good
**Presentation:** 3 good
**Contribution:** 2 fair
**Rating:** 6
**Confidence:** 4

**Summary:**

This paper proposes a dataset of cloth draped on objects intended to learn a model for physics-based upsampling. The central idea of the dataset is to run Progressive Cloth Simulation (PCS) that generates multi-resolution draped cloth with consistent output across different levels. In addition to using the dataset to benchmark different existing upsampling approaches, the paper also proposes a learning-based upsampling approach which is also evaluated on the dataset.

**Strengths:**

- This paper presents a large-scale (over 1M) dataset of consistent draped cloth across different levels of detail. The authors clearly have spent a lot of effort in the design of variation in the dataset, including cloth material, shape, and size, as well as the colliding object.

- The use of the proposed dataset to benchmark existing geometric and physics-based upsampling methods is interesting in demonstrating the output properties of each method in aspects such as geometric feature, intersection, and physical energy.

**Weaknesses:**

- My major concern with the proposed dataset is the limited variety of cloth drape that is covered within it. It is true that the authors have put efforts into ensuring variation in factors such as different material types and collider objects, but it still spans a quite limited space of possible cloth drapes. For example, the dataset only includes cloth of square and round shapes, while in the real world, many interesting cloth examples do not belong to either of them, especially garments draped on human bodies which are of natural importance due to their applications. As another example, the final state of the draped cloth is significantly influenced by its initialization. For example, a cloth initialized as flat will be draped completely differently from being initialized evenly folded. Let alone all the infinite possibilities of manual interaction, for example in Fig. 1 (teaser) of the PCS paper (Zhang et al. 2022).

- Due to the above limitation, I am concerned about the ability of this dataset to benchmark various upsampling algorithms under **complicated** deformation, let alone training machine learning models for complicated deformation. Notice that the perspective of judging a dataset is different from the algorithm that generates it. In the original PCS paper, even though relatively simple cloth shape and initialization have been demonstrated, it is less of a concern because as a physics-based algorithm, we generally have more confidence that it will work similarly for cloth of other shapes and from different initialization. In contrast, as a dataset intended for learning algorithms generated by this algorithm, its range of samples directly determines its scope of applicability because the common sense in machine learning is that a model cannot learn patterns outside its training distribution. In my own experience, the scenario where we are interested in a **square** cloth draped on an object without additional interaction is extremely rare.

 - Although PCS is by itself an excellent algorithm for multi-scale quasistatic cloth simulation, I am wondering to what extent it makes sense to use its output for the evaluation that is conducted in this experiment. On one hand, algorithms such as Loop subdivision or Neural Subdivision (Liu et al. 2020) were designed without physics in consideration at all. On the other hand, PCS is not without its own limitations. For example, the output in different levels still has some noticeable discrepancies, which is most obviously in the upper row of Fig. 9 where the paper cloth may or may not touch the ground depending on its resolution. I am wondering whether the authors have a good response for justification at a methodology level.

**Questions:**

- On page 4, the paper writes that "researchers end up repeatedly and unnecessarily generating data". It may be good to avoid saying "unnecessary" because its tone may sound less respectful to the authors of the above work.
- In Section E of the Supplementary document, it should be good to have a more detailed explanation of the quantities used as metrics and list the citations of their sources. The current form reads like plugging in the name of the term into the template of "we compute [X] between the prediction and the ground truth".
- If I understand correctly, each level of upsampling in the dataset follows the midpoint subdivision as in Loop subdivision. Therefore, is it true that all the methods benchmarked in Table 1 should lead to the same topology? Is it possible to, or how can it be used for upsampling schemes that may not produce the same topologies?
- In Sec. 6.2, it is made clear that the training/test split is performed across cloth material and collision objects. How is the split performed for experiments in Sec 6.1?
- How does the proposed upsampling method compare with the original PCS method in terms of runtime performance etc.? It should be good to provide an analysis.

I am willing to increase the rating if the above concerns are addressed in the rebuttal.

---

> ### Author Response · Authors · 2023-11-17
> **Author response (1/2)**
>
> # General answer for clarifying the big picture:
>
> We appreciate the recognition of the importance of a high-quality multi-resolution, cloth-drape dataset and benchmark evaluation protocols.
>
> Here we'll focus on the important raised question regarding sufficient variety in our dataset:
>
> Of course, we agreed that for the cloth upsampling task (and likewise generally) further increasing variety is always of benefit. Indeed we are excited to increase the dataset variety moving forward. At the same time, we'd argue that datasets are inherently constrained; a benchmark, in practice, can never comprehensively encompass every conceivable scenario within the environment. A dataset, at its core, is a substantial and well-defined collection of data, carefully scoped for the purposes of training and evaluation. The possibility of "better" should not come at the cost of the improvement for SOTA that this dataset provides.
>
> Meanwhile, the natural draping of cloth itself, independent of material, boundary shape, human body draping and seamed garment panels, already produces detailed and complicated deformations. We'd like to disambiguate this important deformation complexity (for which existing learning methods already have significant trouble capturing) from the additional complexities posed by even-richer clothed human scenarios. To be very clear, we do not have confidence nor evidence that any of the evaluated upsampling methods will be applicable in such contexts. However, we want to highlight that **a predicate for new upsampling methods to work in such human-based scenarios is for them to first succeed in the current proposed direct-draping upsampling task for the current benchmark**. As standard in many other grand-challenge learning tasks, we begin with a simpler yet still well-scoped and well-beyond current SOTA benchmark task here. (As we demonstrate in our evaluation, this is a task that current learning-based upsampling methods are still far from able to address, similar to how early image recognition models were tested using MNIST and CIFAR-10 before advancing to more complex ones.) With successful methods developed, we look forward to enhancing the cloth upsampling dataset with even more complex draping and human interaction scenarios. Of course, this comes with significant further investment in time and computation resources. We note that this process is familiar in public datasets like PartNet (Mo et al. 2019) (evolved from ShapeNet (Chang et al. 2015)), Objaverse-XL (Deitke et al. 2023a) (originally Objaverse (Deitke et al. 2023a)), ImageNet (Russakovsky et al. 2015), and PASCAL VOC (Everingham et al. 2009), to name just a few – all of these started with constrained features and underwent significant expansion following their initial publication. Therefore, taking this initial step is not merely important but crucial for continued advancement in the field.

---

> ### Author Response · Authors · 2023-11-17
> **Author response (2/2)**
>
> # Answers to specific questions:
>
> > As another example, the final state of the draped cloth is significantly influenced by its initialization. For example, a cloth initialized as flat will be draped completely differently from being initialized evenly folded.
>
> Yes! To capture a range of initializations, we vary the initial orientation of the cloth by different degrees, which indeed leads to distinct draping behaviors (detailed in Section 4 of our paper). Additionally, we emphasize the absence of a large-scale, high-quality cloth drape dataset and the substantial effort and resources needed to create one. Though initializing the cloth differently, such as by even folding, could further diversify our dataset, we are content with its current state and leave exploiting these variations for future work.
>
> ---
>
> > Although PCS is by itself an excellent algorithm for multi-scale quasistatic cloth simulation, I am wondering to what extent it makes sense to use its output for the evaluation that is conducted in this experiment. On one hand, algorithms such as Loop subdivision or Neural Subdivision (Liu et al. 2020) were designed without physics in consideration at all. On the other hand, PCS is not without its own limitations. For example, the output in different levels still has some noticeable discrepancies, which is most obviously in the upper row of Fig. 9 where the paper cloth may or may not touch the ground depending on its resolution. I am wondering whether the authors have a good response for justification at a methodology level.
>
> PCS coarse-level models (or for that matter any coarse-res proxy cloth model) must choose differing approximation tradeoffs; for example between capturing the general curvature of a higher-fidelity model vs. touching ground at the same corresponding points. Here the dataset we generate with PCS makes various such tradeoffs for each coarse-level model. We certainly agree that it would be interesting to consider other possible alternatives (each with their own pros and cons) for coarse-level geometry but here the key and critical point is that the PCS-generated data provides correspondences across resolutions for these proxies all the way up to a finest high-fidelity high-resolution simulated drape. To our knowledge, this was not available in any previous dataset and is clearly an important feature for any physics-based cloth upsampling benchmark.
>
> ---
>
> > On page 4, the paper writes that "researchers end up repeatedly and unnecessarily generating data". It may be good to avoid saying "unnecessary" because its tone may sound less respectful to the authors of the above work.
>
> We have updated the paper as suggested.
>
> ---
>
> > If I understand correctly, each level of upsampling in the dataset follows the midpoint subdivision as in Loop subdivision. Therefore, is it true that all the methods benchmarked in Table 1 should lead to the same topology?
>
> Yes, each level of upsampling follows the midpoint subdivision as in Loop subdivision. All methods in Table 1 will lead to the same topology (genus).
>
> ---
>
> > Is it possible to, or how can it be used for upsampling schemes that may not produce the same topologies?
>
> Yes! As long as the method's output is given, we can compute the metrics between them.
>
> ---
>
> > In Sec. 6.2, it is made clear that the training/test split is performed across cloth material and collision objects. How is the split performed for experiments in Sec 6.1?
>
> We shuffle the dataset and randomly split the dataset into the training (80%), validation (10%) and test (10%) sets.
>
> ---
>
> > How does the proposed upsampling method compare with the original PCS method in terms of runtime performance etc.? It should be good to provide an analysis.
>
> See the table below. The unit is second.
>
> |              |      PCS | Ours |
> | :--- | ---: | ---: |
> | Level 0  |    16.40 |       - |
> | Level 1  |      7.96 |  0.69 |
> | Level 2  |      6.92 |  0.75 |
> | Level 3  |  281.61 |  0.81 |
> | Level 4  |    18.34 |  1.15 |
> | Others   |    33.37 |  1.43 |

---

> > ### Comment · Reviewer_QcF6 · 2023-11-18
> >
> > Can you add a detailed explanation of evaluation metrics in Sec. E as suggested?

---

> ### Author Response · Authors · 2023-11-18
> **Author response**
>
> Sure! We have included the details of the evaluation metrics and updated the paper. Please see Appendix E for the highlighted text sections. We will release our code for computing these metrics upon acceptance.

---

> > ### Author Response · Authors · 2023-11-23
> >
> > Dear Reviewer QcF6,
> >
> > As the discussion period is coming to an end, we aim to ensure that we have thoroughly addressed all your concerns. Could you kindly confirm if the clarifications provided are sufficient and clear? If there are any outstanding issues or further questions, we welcome you to share them with us.
> >
> > We greatly appreciate your time and input.
> >
> > Best regards, The Authors.

---

### Official Review · Reviewer_L7uF · 2023-10-31

**Soundness:** 2 fair
**Presentation:** 2 fair
**Contribution:** 2 fair
**Rating:** 5
**Confidence:** 3

**Summary:**

This paper presents a multi-resolution dataset of several cloth types developed using state of the art technique, progressive cloth simulation (PCS).  This technique allows a consistent transition from one resolution to another by avoiding self-intersections or artefacts, which are commonly found in other state of the art physics-based upsampling methods.  This dataset is developed by considering the geometry of the collider object into account in order to reduce the cloth-object penetration.


Overall, the paper attempts an important problem but the results are incremental to existing methods. It is not clear that how reliable the dataset generated by the proposed methodology is beyond state of the art upsampling methods.

**Strengths:**

The drapings look realistic and the geometry seems to be consistent across levels. The number of self intersections in table 1 are significantly lower than proposed methods. The performance across other metrics is promising too.
The improvement in the Neural subdivision method (Liu et al, 2020) to take into account material properties of the cloth and geometry of collision objects is a plus.
The computation time reported in table 9 in the paper indicates the computational simplicity of the method.

**Weaknesses:**

The performance degrades sharply with unseen data (table 2 and 3). It is only slightly better than Neural Subdivision, on which this dataset is based. The unseen material is denim, which does not produce much wrinkles and therefore, does not represent a challenging testing scenario.
I am curious how ground truth is obtained. At high resolutions, moth physics based methods are prone to creating self-intersections while capturing details. How is this approach making sure that the ground truth is absolutely correct? The authors should comment on its reliability.
Why is collider object geometry only taken into account as point net features? Why are there no geometric losses to minimise the intersection between the collider object and the cloth? It could have helped better to improve the results. In most applications related to draping, the collider object is known. Using it directly by minimising the cloth-object interaction is widely popular. It is not clear why the authors did not include this and chose to enforce this additionally.

**Questions:**

See weakness section

---

> ### Author Response · Authors · 2023-11-13
> **Request for clarification**
>
> Dear reviewer,
>
> Thank you for your valuable review. As we prepare a full detailed response, we would like to quickly ask for some clarifications:
>
> In the summary section, you commented:
>
> > The paper attempts an important problem but the results are incremental to existing methods. It is not clear that how reliable the dataset generated by the proposed methodology is beyond state of the art upsampling methods.
>
> By saying "the results", are you referring to the multi-resolution mesh tuples generated by PCS (Zhang et al., 2022) or the predicted upsampled cloth results made by our proposed learning-based baseline method?
>
> Correspondingly, does "the proposed methodology" mean the PCS method (Zhang et al., 2022) or our proposed learning-based baseline method for cloth upsampling?
>
> ---
>
> In the strength section, you commented:
>
> > The number of self intersections in table 1 are significantly lower than proposed methods.
>
> Which methods do "proposed methods" here refer to?
>
> Thanks,
>
> Authors

---

> > ### Comment · Reviewer_L7uF · 2023-11-14
> >
> > In the summary section, you commented:
> >
> > The paper attempts an important problem but the results are incremental to existing methods. It is not clear that how reliable the dataset generated by the proposed methodology is beyond state of the art upsampling methods.
> >
> > By saying "the results", are you referring to the multi-resolution mesh tuples generated by PCS (Zhang et al., 2022) or the predicted upsampled cloth results made by our proposed learning-based baseline method?
> >
> > -->  by results, I refer to the predicted upsampled cloth results made by your proposed learning-based baseline method.
> >
> > Correspondingly, does "the proposed methodology" mean the PCS method (Zhang et al., 2022) or our proposed learning-based baseline method for cloth upsampling?
> >
> > -->  by proposed methodology in summary, I refer to the your proposed learning-based pipeline that uses PCS.
> >
> > In the strength section, you commented:
> >
> > The number of self intersections in table 1 are significantly lower than proposed methods.
> >
> > Which methods do "proposed methods" here refer to?
> >
> > -->  There is a typo here. The number of self intersections in table 1 are significantly lower than 'existing' methods.
> >
> >
> > In general, the proposed dataset is built using PCS in a collision-aware context, which is learnt from collider geometries. Since a small number of collider geometries are considered, one cannot guarantee a collision awareness wrt generic geometries. This is why the number of self-interactions (and other metrics) on unseen data (table 2, 3) are much higher than the one in table 1.

---

> ### Author Response · Authors · 2023-11-17
> **Author response (1/3)**
>
> # General answer for clarifying the big picture:
>
> We appreciate the reviewer's acknowledgment of the importance of the physics-based cloth upsampling task in both simulation and learning communities. However, to ensure a fair assessment of our paper’s contributions, we respectfully highlight certain factual misunderstandings in the review:
>
> ---
>
> > In general, the proposed dataset is built using PCS in a collision-aware context, which is learned from collider geometries.
>
> To clarify, our dataset is generated by the PCS algorithm (Zhang et al. 2022) powered by C-IPC (Li et al. 2021), a state-of-the-art cloth simulator with well-evaluated physical accuracy in cloth draping accuracy, contact-force resolution and ensuring intersection-free results. Specifically, the dataset is **not** generated with the application of any learning components. Please see the abstract, introduction, background sections and appendix A of our paper. It seems there might be a significant mix-up by the reviewer, confusing the data generation process with our proposed learning-based baseline method for cloth upsampling.
>
> To enhance clarity, our paper is structured around two main components: (1) **Data generation** using PCS, and (2) **Benchmarking** existing upsampling methods for cloth upsampling (detail enhancement) using the generated dataset.
>
> - For (1), we employ PCS to create a high-quality dataset, as detailed in the third paragraph of the introduction section, where PCS's effectiveness is discussed in detail.
>
> - For (2), we benchmark five existing upsampling methods and further introduce a new learning-based baseline method for cloth upsampling to foster future research. All these six methods have access to and are tested on the same PCS-generated dataset, enabling a fair comparison of their cloth upsampling performance in a unified testbed.
>
> ---
>
> > This dataset is developed by considering the geometry of the collider object into account in order to reduce the cloth-object penetration.
>
> Same as above. We apply PCS to generate data. The data generated by PCS is intersection- and penetration-free.
>
> ---
>
> > Since a small number of collider geometries are considered, one cannot guarantee a collision awareness wrt generic geometries. This is why the number of self-interactions (and other metrics) on unseen data (table 2, 3) is much higher than the one in Table 1.
>
> We respectfully disagree. In our network design, we use the PointNet encoder from [Mescheder et al. 2019] which has been pre-trained on a large collection of shapes for learning occupancy fields (see Appendix F for details). This network design allows our network to be aware of the collider object geometry.
>
> We note that it is expected that a trained network performs better on in-distribution data (in our case, seen collider objects as in Table 1) and has performance degradation on out-of-distribution data (in our case, unseen collider objects as in Table 3). As shown in Tables 2 and 3, while all methods suffer from performance drops on unseen data (materials and collider objects), our additional new learning-based baseline method still outperforms all competing methods by a large margin (at least 13% improvement on the Euclidean distance metric $d_\text{corr}$ over U-DNN and 18% over Neural Subdivision).
>
> ---
>
> > The computation time reported in Table 9 in the paper indicates the computational simplicity of the method.
>
> We respectfully disagree. The runtime in Table 9 refers to that of the PCS method (i.e., data generation), and not to our proposed learning-based baseline method for cloth upsampling.

---

> ### Author Response · Authors · 2023-11-17
> **Author response (2/3)**
>
> Considering the misunderstandings identified, we note with concern that the majority of the review’s focus—addressing **both strengths and weaknesses**—seems to concentrate predominantly on the performance of our newly proposed baseline method. We would like to clarify that **the newly proposed baseline method is not the central emphasis of our work**. Our paper's primary contributions, as a **dataset and benchmark paper** (marked in the primary area section), lie in generating a large-scale, high-quality dataset of cloth drapes (currently **no similar public datasets exist**), benchmarking existing upsampling methods on our dataset using the same dataset, and developing evaluation protocols for future research. Moreover, our detailed analysis of existing upsampling methods has unveiled a new crucial insight: the diversity and complexity of the cloth drapes in our dataset expose severe limitations in existing upsampling methods (See Table 1 and Figures 4, 7 and 8). To demonstrate the potential of addressing this gap, we further develop a learning-based baseline method for cloth upsampling that improves existing upsampling methods. This addition, though not claimed as a novelty by us, serves as a valuable contribution we offer. As shown in Figures 4, 7 and 8, none of the existing upsampling methods can predict reasonable results. Our simple yet effective approach of incorporating collision geometries and material properties into the neural network has yielded plausible outcomes featuring detailed wrinkling and fewer intersections. We aspire that this insight serves as an inspiration for future research, encouraging deeper exploration and further advancements in the field.
>
> Given the substantial misunderstandings that have diverged from the principal focus of our paper, we kindly request the reviewer to re-evaluate our submission in light of the clarifications provided.

---

> ### Author Response · Authors · 2023-11-17
> **Author response (3/3)**
>
> # Answers to specific questions:
>
> > The performance degrades sharply with unseen data (Tables 2 and 3). It is only slightly better than Neural Subdivision, on which this dataset is based.
>
> We respectfully disagree. Quantitatively, as shown in Tables 2 and 3, our baseline method improves in performance over Neural Subdivision by over 18% in the Euclidean correspondence metric $d_\text{corr}$. Qualitatively, as shown in Figures 4, 7 and 8, Neural Subdivision fails to predict wrinkles, whereas our baseline method predicts plausible cloth drapes with intricate details.
>
> We reiterate that as a dataset and benchmark paper, **the newly proposed baseline method is not the central emphasis of our work** – our baseline method serves as an example to showcase potential improvements that can be realized over existing techniques when analysis is performed with the new benchmark. We look forward to future research building on the foundations of both the benchmark and this new baseline to further enhance physics-based upsampling performance and quality.
>
> ---
>
> > The unseen material is denim, which does not produce many wrinkles and therefore, does not represent a challenging testing scenario.
>
> We respectfully disagree. As shown in Figure 2 and more examples in Figure 1, the denim material produces many intricate wrinkles (with very thin thickness) and thus represents a challenging testing scenario.
>
> ---
>
> > I am curious how ground truth is obtained.
>
> As mentioned in the third paragraph of the introduction section, we use PCS (Zhang et al., 2022) to generate a multi-resolution tuple of corresponding cloth drapes. In each tuple, we treat the lowest-resolution cloth drape as input and the other cloth drapes as the ground truth for the predictions of subsequent resolution levels.
>
> ---
>
> > Why is collider object geometry only taken into account as point net features?
>
> Our goal is to let our network be aware of the collider object geometry. As mentioned in Appendix F, we use the pre-trained PointNet encoder from [Mescheder et al. 2019] which has been pre-trained on a large collection of shapes for learning occupancy fields. This allows the PointNet encoder to have a good understanding of the geometry of the input point cloud. While there are certainly other possible design choices, our ablation study experiment in Section 6.3 shows that such a network design allows our network to be aware of the collider object geometry, resulting in significant performance improvement.
>
> ---
>
> > Why are there no geometric losses to minimize the intersection between the collider object and the cloth? It could have helped better to improve the results. In most applications related to draping, the collider object is known. Using it directly by minimizing the cloth-object interaction is widely popular. It is not clear why the authors did not include this and chose to enforce this additionally.
>
> We would like to reiterate that in this paper our focus is on collecting a large-scale, high-quality dataset of cloth drapes, benchmarking existing upsampling methods on our dataset, and developing evaluation protocols for future research. Our results show the limitations of existing upsampling methods. We provide an additional learning-based baseline method for cloth upsampling that improves existing upsampling methods, and we show that such a simple network design results in significant performance improvement. We agree that having geometric losses such as the collision penalty in SNUG (Santesteban et al. 2022) to minimize the collider-cloth intersection could potentially help improve performance. However, given that the proposed learning-based baseline method is not the central focus of our work, we leave it as interesting future work.

---

> > ### Author Response · Authors · 2023-11-23
> >
> > Dear Reviewer L7uF,
> >
> > As the discussion period is coming to an end, we aim to ensure that we have thoroughly addressed all your concerns. Could you kindly confirm if the clarifications provided are sufficient and clear? If there are any outstanding issues or further questions, we welcome you to share them with us.
> >
> > We greatly appreciate your time and input.
> >
> > Best regards,
> > The Authors.

---

### Author Response · Authors · 2023-11-20
**Author/reviewer disucssion**

Dear Reviewers,

Thank you for providing insightful feedback to help us improve our work. We have addressed all the questions in detail and revised the manuscript. Please have a look at our response to each question and the revised manuscript. Please let us know if there is anything that needs further clarification. Thank you.

Authors

---

### Meta-Review · Area_Chair_VEq4 · 2023-12-06

**Metareview:**

The paper describes a new dataset of one million cloth meshes in contact with collider objects, that can be used to train upsampling methods for cloth simulation.

Strengths:
 - The dataset is large, and its use in training the authors' proposed new method demonstrates its utility in this field

Weaknesses:
 - Reviewer qJGw clearly states in their opening review that they understand that the purpose of the dataset is in upsampling.  However, their assessment of the value of the dataset is reduced because in their view "the significant shift" in approach of the field.  Their assessment that the lack of dynamic (and non-resting, and non-gravity) data may affect the utility of the dataset stands, even when viewed as an upsampling dataset - it is not clear that moving cloth obeys the same upsampling rules as static cloth.

More generally, this topic is peripheral for ICLR (the paper's own many references include only a few machine learning venues, citing the generic concepts such as the Adam optimizer and PyTorch).  Given the reservations of the reviewers, who are chosen as being in the field of the paper, its relevance to a wider ICLR audience is reduced.

**Justification For Why Not Higher Score:**

Peripheral content (applied machine learning is of course in scope for ICLR, but only when it has aspects that permeate to more general machine learning).

**Justification For Why Not Lower Score:**

n/a

---

### Decision · Program_Chairs · 2024-01-16

Reject